# Changes in ground deformation prior to and following a large urban landslide in La Paz, Bolivia, revealed by advanced InSAR

Nicholas J. Roberts[1], Bernhard T. Rabus[2], John J. Clague[1], Reginald L. Hermanns[3,4], Marco-Antonio Guzmán[5], Estela Minaya[6]

[1]Department of Earth Sciences, Simon Fraser University, 8888 University Drive, Burnaby, Canada, V5A 1S6
[2]School of Engineering Science, Simon Fraser University, 8888 University Drive, Burnaby, Canada, V5A 1S6
[3]Geological Survey of Norway, P.O. Box 6315 Sluppen, Trondheim, Norway, 7490
[4]Department of Geoscience and Petroleum, Norwegian University of Science and Technology, Trondheim, Norway, 7491
[5] Instituto de Investigaciones Geológicas, Universidad Mayor de San Andrés, Pabellon 3, Campus Universitario Cota Cota, La Paz, Bolivia, 35140
[6]Observatorio San Calixto, Indaburo 944, La Paz, Bolivia, 12656

*Correspondence to*: Nicholas J. Roberts (nickr@sfu.ca)

**Abstract.** We characterize and compare creep preceding and following the complex 2011 Pampahasi landslide (~40 Mm$^3$ ± 50%) in the city of La Paz, Bolivia, using spaceborne RADAR interferometry (InSAR) that combines displacement records from both distributed and point scatterers. The failure remobilised deposits of an ancient complex landslide in weakly cemented, predominantly fine-grained sediments and affected ~1.5 km$^2$ of suburban development. During the 30 months preceding failure, about half of the toe area was creeping at 3-8 cm/a and localized parts of the scarp area showed displacements of up to 14 cm/a. Changes in deformation in the 10 months following the landslide demonstrate an increase in slope activity and indicate that stress redistribution resulting from the discrete failure decreased stability of parts of the slope. During that period, most of the landslide toe and areas near the headscarp accelerated, respectively, to 4-14 and 14 cm/a. The extent of deformation increased to cover most, or probably all, of the 2011 landslide as well as adjacent parts of the slope and plateau above. The InSAR-measured displacement patterns, supplemented by field observations and optical satellite images, reveal complex slope activity; kinematically complex, steady-state creep along pre-existing sliding surfaces accelerated in response to heavy rainfall, after which slightly faster and expanded steady creeping was re-established. This case study demonstrates that high-quality ground-surface motion fields derived using spaceborne InSAR can help to characterize creep mechanisms, quantify spatial and temporal patterns of slope activity, and identify isolated small-scale instabilities; such details are especially useful where knowledge of landslide extent and activity is limited. Characterizing slope activity before, during, and after the 2011 Pampahasi landslide is particularly important for understanding landslide hazard in La Paz, half of which is underlain by similar large paleolandslides.

## 1 Introduction

Creep – steady, imperceptibly slow movement under the influence of gravity – precedes many large landslides in both bedrock (e.g. Chigira and Kiho, 1994; Kilburn and Petley, 2003) and soil (e.g. Kalaugher et al., 2000; Petley et al., 2002). Its temporal patterns can improve understanding of slope-deformation processes (Petley et al., 2002) and the possible timing of impending catastrophic failure (Voight, 1989; Crosta and Agliardi, 2003; Federico et al., 2015). Its spatial patterns indicate the style (Gischig et al., 2011; Schlögel et al., 2016) and approximate magnitude of instability. Consequently, characterization of temporal and spatial patterns of slow deformation is a focus of many landslide hazard assessments, monitoring systems, and warning programs (Eberhardt, 2012; Baron and Supper, 2013; Hermanns et al., 2013a). In many cases, however, pre-failure creep is not quantified, either because it is not recognized prior to catastrophic failure or because failure carries a relatively low risk to people and property.

Creep is rarely monitored following a landslide, except at sites where renewed catastrophic failure is expected (e.g. Gischig et al., 2011). Limited post-event monitoring sometimes stems from an assumption that stress release during catastrophic failure enhances slope stability. In addition, instrumentation that may have been present on or within the slope is typically destroyed during failure (e.g. Crosta and Agliardi, 2003). Consequently, activity soon after failure is generally unclear and can rarely be compared with pre-failure deformation. Such records are, however, of utmost importance in light of observed spatiotemporal clustering of failure events (e.g. Hermanns et al., 2006, and references therein; Crosta et al., 2017; Hilger et al., 2018).

Spaceborne RADAR interferometry (InSAR) has long been recognized for its promise in characterizing the spatiotemporal evolution of slopes (e.g. Petley et al., 2002; Colesanti and Wasowski, 2006; Wasowski and Bovenga, 2014). Continued technological advancements and data availability offer the potential for further improvements to the characterization of slope behaviour (Wasowski and Bovenga, 2014), including prediction of future failure (Moretto et al., 2017). Improvement of RADAR systems is increasing ground resolution and phase quality, and the establishment of satellite constellations is improving temporal resolution and interferometric coherence. Due to such improvements, pre-failure accelerated creep has recently been detected and measured, both for natural (Intrieri et al., 2018; Handwerger et al., 2019) and cut (Carlà et al., 2018) slopes. Such advancements similarly improve opportunities to analyze post-failure creep.

Our quantification of creep during the 30 months preceding and 10 months following the largest modern landslide in the city of La Paz, Bolivia (2011 Pampahasi landslide, ~40 Mm$^3$) using InSAR demonstrates complex slope behaviour. Contrary to the sometimes-invoked view that a major failure reduces large-scale instability, the affected slope shows enhanced post-failure activity, highlighting the complexity of slopes stability in La Paz. The Pampahasi case study illustrates the necessity for land-use planning views that better align with the complexity and commonly recurrent nature of large landslides. The

InSAR technique we employ is optimized to maximize displacement record density and distinguish abrupt changes in motion, which together improve the characterization of spatially complex deformation patterns typical of landslides. The resulting displacement fields provide nearly spatially continuous quantification of motion over large portions of the landslide and surrounding terrain that enable discrimination of localized motion and aid interpretation of failure mechanisms.

## 2 Setting and material properties

### 2.1 La Paz region

A tributary of the Amazon River system penetrates the Cordillera Real ~60 km southeast of La Paz and drains the eastern margin of the otherwise internally drained Altiplano Plateau (~4000 m a.s.l.) (Fig. 1A). The city of La Paz (population ~0.8 million) is located within the upper part of this watershed where the valley system has incised into up to 800 m of weakly lithified Neogene and Pleistocene sediments shed from the Cordillera Real (Dobrovolny, 1962) (Fig. 1B). South of the city, incision has exposed folded and faulted, Paleozoic to Mesozoic metasedimentary basement rocks forming the root of the range (Fig. 1B). The overlying Cenozoic fill sequence comprises over 600 m of predominantly fine-grained lacustrine and fluvial sediments of the weakly lithified Mio-Pliocene La Paz Formation. It coarsens upward and toward the cordillera into more competent Late Pliocene and Early Pleistocene glacial and proglacial sediments (Roberts et al., 2018) that are up to 400 m thick. The fill has been offset by numerous faults trending northwest and west (Dobrovlny 1962; Lavenu, 1978; Roberts et al., 2018) (Fig. 1B).

The slopes and floors of the La Paz valley system are mantled by colluvium, including deposits of large (up to 50 km$^2$; Dobrovolny, 1968) paleolandslides derived predominantly from the La Paz Formation (Dobrovolny, 1962; Anzoleaga et al., 1977) (Fig. 1B). Most of the paleolandslides date to the Late Pleistocene and Holocene (Dobrovolny, 1962; Hermanns et al., 2012). Historic instability is concentrated within paleolandslide deposits (e.g. Fig. S1), particularly at their margins, likely reflecting low residual strengths and possibly differential stresses related to suspected ongoing creep. Failures <1 Mm$^3$ in size occur yearly and happen mainly during the rainy season (December-March) (O'Hare and Rivas, 2005; Roberts, 2016). In contrast, of the seven historic landslides larger than 1 Mm$^3$, four happened during the dry season (April-November; Table S1). Although general geomorphic and geologic characterization has been undertaken for some of the larger landslides (Dobrovolny, 1962; Anzoleaga et al., 1977; Quenta et al., 2007, 2008; Hermanns et al., 2012), detailed site investigations are lacking. Land-use decisions in La Paz have historically overlooked recurrent slope failures, with large landslide complexes being either unrecognized or repeatedly resettled after reactivations.

## 2.2 Pampahasi area

Pampahasi Plateau is 2.2 km long, 0.2-0.7 km wide, and forms part of the interfluve between ríos Orkojahuira and Irpavi (Fig. 1B). Its margins are generally scalloped (Fig. 2) and in several places terminate in nearly vertical cliffs dropping 20-80 m to irregular, locally steep valley slopes below. The plateau surface dips slightly southward and lies at an elevation several hundred metres below the local southwest-dipping Altiplano and its erosional remnants (Fig. 3A).

The Río Irpavi valley is cut into well stratified, fine-grained facies of the La Paz Formation, which comprises horizontal to sub-horizontal silt and sand beds interlensing with mainly granitic pebble-cobble gravel (Fig. 3B). The variable texture and limited (<1 km) lateral continuity of these beds are typical of the unit, particularly its middle part (cf. Ahlfeld, 1945a; Dobrovolny, 1962; Bles et al., 1977). The age of the La Paz Formation beneath Pampahasi Plateau (Fig. 3B) is constrained by the 5.4-Ma Cotacota Tuff (Lavenu et al., 1989; Servant et al., 1989), which outcrops at ~3520 m a.s.l. in the southern part of the city (Lavenu et al., 1989), and the 2.74-Ma Chijini Tuff (Roberts et al., 2017), which crops out along the west margin of the plateau (Dobrovolny, 1962) between 3800 and 3850 m a.s.l. The local Altiplano surface formed on top of the thick sequence of Plio-Pleistocene glacial sediments by 1.8 Ma or possibly as recently as 1.0 Ma (Roberts et al., 2018), attaining an elevation of approximately 4200 m a.s.l. in the vicinity of Pampahasi (Fig. 3A).

About 400 m of incision of the Altiplano surface by ancestral Río Irpavi after 1.8-1.0 Ma formed a south-sloping (Dobrovolny, 1962), valley-bottom unconformity onto which the Middle Pleistocene Pampahasi gravel (Fig. 3B) was deposited, either as fluvial channel deposits (Ahlfeld, 1945b) or as an ancient landslide (Bles et al., 1977). This multilithic silty sandy pebble-cobble gravel is commonly more than 20 m thick (Dobrovolny, 1962), and may be up to 50 m thick along the plateau's southern edge where the unit overlies a locally steep erosional contact (Dobrovolny, 1955).

Many slopes descending from Pampahasi Plateau are mantled by ancient landslide deposits derived from the La Paz Formation (Dobrovolny, 1962; Anzoleaga et al., 1977). The deposits consist of largely intact brittle blocks of La Paz Formation within a matrix of softer, ductile and completely disaggregated La Paz Formation. Small (<1 to 20 ha) failures evident in aerial photographs have occurred since at least the early twentieth century in many places within large (~20-200 ha) prehistoric failures. The slope descending ~380 m to Río Irpavi directly east of the southern half of Pampahasi Plateau comprises the undated Pampahasi paleolandslide (Fig. 2A).

Details of the prehistoric Pampahasi paleolandslide, which substantially predates historic records, must be inferred from its surface expression because no subsurface investigations have been conducted on this slope. Its morphology indicates a complex failure, but whether it represents one or multiple events is uncertain. Rotational displacement of the upper quarter of the slope transitions downward to eastward sliding and localized flow. The landslide displaced Río Ipravi eastward from its alignment farther upstream and downstream by at least 300 m. The substantial height of the headscarp (up to 80 m) suggests

that the failure zone is many tens of metres below surface in the upper part of the landslide. The failure depth lower on the landslide is likely much shallower.

Surface incision of the Pampahasi paleolandslide by several gullies suggests that it is of substantial antiquity. The adjacent and similarly incised Villa Salomé paleolandslide is likely a separate landslide complex because its failure surface is higher and is separated from the Pampahasi paleolandslide, along Río Jankopampa by a >120-m-thick sequence of intact La Paz Formation (Fig. 2A). In the area of the Pampahasi paleolandslide, the Río Irpavi channel is modified with check dams and concrete armouring. Before the 2011 failure, western tributaries of Río Irpavi draining the Pamaphasi and Villa Salomé paleolandslides were little modified and had incised into weak loose material. The main channels draining the area (ríos Chujilluncani and Jankopampa, Fig. 2) have since been entombed in concrete culverts. In light of the geomorphic evidence of recurrent instability at the sites, Scanvic and Girault (1989) recommended that this area not be developed. However, the initially sparse development greatly expanded during the last decade of the twentieth century and first decade of the twenty-first, resulting in the establishment of several large neighbourhoods.

## 2.3 Geomechanical properties

The different characteristics of the Pampahasi gravel and La Paz Formation give rise to contrasting material properties, which have been documented by Anzoleaga et al. (1977; summarized in Table S2). The Pampahasi gravel (50-75% clasts, 19-50% sand and silt, <6% clay) is permeable (k ~1 * $10^{-3}$ m/s), has low to medium plasticity (plasticity index ≤10), and exhibits high internal friction (>30°), particularly when dry. Fine-grained zones are weaker, but uncommon.

The properties of the La Paz Formation are variable due to the unit's heterogeneity. Coarse sand and gravel zones (35-75% clasts, 13-65% sand and silt, <12% clay) are of similar permeability to the Pampahasi gravel, but are commonly cemented by carbonate (≤15%) or iron oxide. Fine-grained zones (≤10% clasts, 40-90% sand and silt, 10-40% clay) are much weaker and, as the main facies, dominate the unit's behaviour. They are weakly cemented by carbonate (<5%) and locally abundant clay. The clay is largely non-expansive, but montmorillonite (smectite) is common in some beds and lenses. Due to low permeability (k <1 * $10^{-5}$ m/s), water circulation is mainly through fractures in interstratified gravel lenses or along joints. Plasticity is medium to high (plasticity index of 7-32), and shear strength ranges from 18° to 37°. The unit is generally strong when dry and undisturbed (cohesion 0.2-0.7 MPa) and stands in near-vertical slopes. However, wet silty sediments have much lower cohesion (0.01-0.1 MPa) and shear strength. Smectite-rich clay lenses have greater cohesion (≥0.1 MPa), but high plasticity (plasticity index of 60-80) and low shear strength (≤14°). Parts of the La Paz Formation that have failed in the past are even weaker due to loss of compaction and localized reduction of shear strength to residual levels (as low as 13° in silty/sandy zones and less 8° or less in clay zones).

## 3 Methods

### 3.1 Failure event characterization

Our characterization of the 2011 landslide is based on field visits in the years before and after the event and on comparison of high-resolution optical satellite images acquired shortly before and after the event. Field observations by municipal staff and eyewitness accounts of residents provide details on the location, type, and magnitude of damage shortly before, during, and after the 2011 failure. We mapped features of the 2011 (26 February to 1 March) landslide from the first cloud-free, post-failure imagery (WorldView-2 acquired on 23 March 2011), and quantified horizontal displacement vectors during the failure by comparing it with a pre-failure (05 January 2011) image from the same sensor, which we coregistered using points outside the landslide-affected area. Due to errors resulting from the lack of orthorectification and from elevation differences cause by the landslide, the estimated horizontal vectors are approximate (rounded to the nearest 5-m increment).

### 3.2 HDS-InSAR

We applied Homogeneous Distributed Scatterer InSAR (HDS-InSAR; Eppler and Rabus, 2011; Rabus et al., 2012) to a stack of 44 Fine Beam mode (5.1 m azimuth x 8.0 m ground-range pixel spacing) RADARSAT-2 ascending scenes (look direction: 36.3° incidence angle from vertical at scene center, 76.0° aspect angle counter clockwise from north) acquired between September 2008 and December 2011 (Supplement, Tables S4 and S5). We processed the scenes separately as a pre-failure stack (32 scenes) and a post-failure stack (12 scenes) to compare deformation patterns before and after the landslide and to reduce decorrelation caused by landslide-induced terrain changes. Although a range of displacement directions is possible, ground-surface displacement vectors due to mass movements are most probable in the downslope direction; we thus estimated true displacement rates from the angular deviation between down-slope vectors and the satellite's line-of-sight (LOS).

The HDS approach combines the strengths of Persistent Scatterer InSAR (PS-InSAR; Ferretti et al., 2001) and Small Baseline Subset InSAR (SBAS-InSAR; Berardino et al., 2002) for considering, respectively, point and distributed targets. A continuously weighted, spatially adaptive filter defines spectrally similar pixel clusters (HDS neighbourhoods) within a rectangular search area based on similarity of their backscatter amplitude time series. The filter is applied to interferograms to generate differential phase and coherence for each HDS neighbourhood (Eppler and Rabus, 2011), which comprises either a single-pixel cluster (a persistent point target) or the weighted mean of a multi-pixel cluster (interpreted as the area covered by a homogenous distributed target). To more efficiently represent the high density of displacement records, we interpolated linear deformation maps from HDS to provide base images of long-term, average LOS deformation over the entire area.

Like PS-InSAR, HDS-InSAR generates high-quality displacement time histories attributable to single pixels, each containing a dominating point target with near-linear displacement characteristics (Colesanti and Wasowski, 2006).

Additionally, it provides time histories for multi-pixel clusters representing homogeneous distributed scatterers. Like SBAS-InSAR, HDS-InSAR characterizes strongly non-linear motion (Necsoiu et al., 2014) using the higher continuity coverage (Lauknes et al., 2010) of distributed targets. Unlike SBAS-InSAR, however, it does not significantly diminish phase quality or reduce spatial resolution. Two key differences set HDS-InSAR and some other advanced InSAR algorithms (e.g. Ferretti

et al., 2011) apart from the PS-InSAR and SBAS-InSAR techniques: the use of adaptive filtering to preserve, as much as possible, spatial resolution while suppressing the noise from surface decorrelation of nearby incoherent pixels; and optimization to characterize spatially uncorrelated ground motion. These features make HDS-InSAR particularly well suited for characterizing landslides, which commonly display ground motion variability over short distances or across abrupt transitions. Although HDS-InSAR and SqueezSAR (Ferretti et al., 2011) share similar adaptive filtering methods, the latter

uses wrapped phase triangulation to invert the interferometric network. In contrast, HDS-InSAR relies on prior unwrapping of the network interferograms, with potential unwrapping errors being corrected iteratively using the network redundancy a-posteriori. Despite the differences in their algorithms, the final accuracy and spatial detail of both advanced InSAR methods are similar.

## 4 2011 Pampahasi landslide

The Pampahasi landslide[1] occurred at the end of February 2011 following a week of variable precipitation (0-39.2 mm daily, Fig. 4), which included one of the wettest days on record (25 February, Table S3). The ~1.5-km$^2$ failure remobilized a large part of the Pampahasi paleolandslide deposit between the east side of Pampahasi Plateau and Río Irpavi (Fig. 2). In addition to previously failed material, it included small volumes of in situ, weakly cemented La Paz Formation and, in the headscarp area, the overlying uncemented Pampahasi gravel (Fig. 2A). None of the sediments involved were substantially weathered.

Several locations in the upper part of the slope showed field evidence of slow ground deformation, largely visible as offset walls and road surfaces (Fig. 5, locations in Fig. 2A) in the years prior to catastrophic failure (Quenta and Calle, 2005; Hermanns et al., 2012). These features were recognized as evidence of an impending landslide (Quenta and Calle, 2005), although not of the magnitude of the 2011 failure. In response, the Municipality of La Paz increased risk communication and installed concrete pillars to remediate what became the 2011 landslide headscarp (Hermanns et al., 2012). However, due to

their localized and shallow nature, the stabilization efforts probably had little, if any influence on the stability of the slope or behaviour of the subsequent failure.

---

[1] The 2011 failure is also locally called the Callapa landslide after one of the neighbourhoods it destroyed.

The 2011 Pampahasi landslide occurred in several phases from the evening of 26 February to 1 March (Fig. 6, locations in Fig. 2B). Eyewitness accounts suggest that failure initiated in an area of earlier small slumps along Río Chujilluncani[2] (Hermanns et al., 2012; Fig. 5A) ~400 m downslope of the east margin of Pampahasi Plateau. Motion was fastest (up to several metres per second) and largely vertical in the head region, which failed shortly after, lasting one to two hours and forming a ~60°, 80-m-high headscarp (Fig. 6A). Movement in the central zone and at the toe, which began soon after the initial formation of the headscarp, was slower (metres per minute to metres per day) and largely horizontal. The degree of ground disturbance and the magnitude of displacement generally decreased to the east and north (Fig. 2B). The northern part of the landslide's toe experienced relatively limited, but economically costly, ground disturbance (e.g. Fig. 6B-D) and moved as a more-or-less coherent mass ~15 m southeastward (Fig. 2B). Deformation of the lower part of the landslide continued for days. A bridge across Río Irpavi (Fig. 6E) was shifted 10 m and uplifted on the morning of 27 February; uplift of several metres at the east end of the bridge suggests a rotational failure zone passing, at least locally, under Río Irpavi. Buildings near the south toe of the landslide (Fig. 6F) collapsed on 28 February and 1 March due to delayed, relatively minor translation.

The surface morphology of the landslide likely reflects a complex failure zone with both rotational and translational components. Assuming a minimum average thickness of the landslide deposit of 15 m, its volume is at least 20 $Mm^3$. However, the volume is more likely about 40 $Mm^3$, given deeper failure in both the upper rotational part (40 m average depth over 32 ha) and the main, predominantly translational body (25 m average depth over 110 ha).

Due to prompt evacuation of the headscarp area and the dominantly moderate to low movement velocities farther down slope, no lives were lost during the landslide. However, about 1000 homes were destroyed and 6000 people were displaced. Rupture of a water line crossing the upper part of the landslide (Fig. 6G) left between 200,000 and 300,000 people in southern La Paz without potable water for several months (Hermanns et al., 2012; Aguilar, 2013). Continued episodic collapse of the steep headscarp in the years since the landslide has resulted in additional loss of homes and repeated expansion of an evacuation zone on Pampahasi Plateau. Large portions of the landslide complex have been recently resettled or are being prepared for reoccupation, with limited control of slope infiltration and runoff.

## 5 Deformation before and after failure

Due to the high spatial density of displacement records, particularly in high-coherence areas (Fig. 7B, C), interpolated linear deformation maps are preferable to HDS point data for interpreting the spatial variability of slope creep (cf. Fig. 7E, F).

---

[2] This stream is sometimes also referred to as Arroyo Pampahasi.

Furthermore, the interpolation weighting suppresses very localized HDS clusters that differ greatly from the average (Supplement, section 4.7). Consequently, these small-scale variations, whether representing surficial movement or noise from uncorrected phase unwrapping errors, are insignificant to large-scale patterns described below. The displacement maps record deep, spatially regular slope movements as well as shallower, more variable movements, but their differentiation requires consideration of displacement patterns and may not always be clear.

InSAR-measured ground deformation in the Pampahasi area is almost entirely restricted to mapped prehistoric landslide deposits, namely the Pampahasi and Villa Salomé paleolandslides (Fig 2A), and the terrain directly behind their headscarps (Fig. 8). Their characteristic displacement histories are shown in Fig. 9 (locations 'i'-'x'). Data gaps in each stack are locations of low coherence that mainly result from decorrelation related to remedial earthworks performed after a 10-ha landslide in 2009 (Fig. 8A, B) and the 2011 landslide (Fig. 8C, D), or from aliasing where displacement rates are greater than the detection threshold of RADARSAT-2 (~2.8 cm LOS [equivalent to a two-way travel distance comparable to RADARSAT-2's 5.6-cm wavelength] over 24 days). Motion toward the satellite suggested by the data in the east-facing slopes (Fig. 8) is improbable, given that it implies uplift or motion upslope. Such issues are almost entirely restricted to the post-failure records (Fig. 8C), suggesting that the small size and limited redundancy of the interferometric network in the thinner stack has resulted in incomplete network correction of either phase unwrapping errors or atmospheric effects.

## 5.1 Pre-failure creep

Prior to the end of February 2011, about 180 ha of the lower part of the slope within the limits of the 2011 landslide was moving at rates as high as 14 cm/a (9 cm/a LOS; Fig. 8A, B). Motion was most widespread over the northern part of the toe of the Pampahasi paleolandslide (2-5 cm/a LOS, 'i'), where it extended into Río Jankopampa alluvium mantling the paleolandslide deposit. There it terminated abruptly ~50 m north of the river (Fig. 8A) at a locally stable slope comprising undisturbed La Paz Formation (Fig. 2). A 1-ha zone directly west of Río Irpavi and halfway along the toe of the paleolandslide ('ii') crept even faster (5-8 cm/a LOS, Fig. 8B). In contrast, activity on the southern part of the paleolandslide toe ('iii') was limited to a few small areas creeping at less than 0.5 cm/a LOS.

Pre-failure movement in the upper half of the Pampahasi paleolandslide was restricted to an area of no more than ~250 m along slope by ~300 m down slope. Given the presence of stationary ground in some parts of the upper half of the landslide (north and south of 'b' in Fig. 8A, B), creep may have been localized to a few smaller areas there. Minor isolated movement (<1 cm/a LOS) is apparent along what became the 2011 landslide's upper lateral margins (e.g. 'iv'). Creep within the Pampahasi paleolandslide deposit beyond the limits of the 2011 failure also was localized and generally slower than 1 cm/a LOS (Fig. 8A). The largest of these areas involved 12 ha of paleolandslide material between the 2009 and 2011 landslides ('v') directly upslope (northwest) of the most active part of the slope toe, and crept at 0.5 cm/a LOS. The most rapid pre-

failure motion (9 cm/a LOS) was restricted to the upper portion of the future landslide where infrastructure damage had been documented in prior years (Fig. 5B, C). Aside from a small, slow-moving (0.5 cm/a LOS; 'vi') area, no motion is evident along the future landslide headscarp, including sites of localized deformation observed in 2005 (Fig. 5D, E) or on the plateau behind it ('vii'-'x').

## 5.2 Post-failure creep

During the 10 months following the landslide, the area of movement on Pampahasi Plateau and the valley slope between it and Río Irpavi increased by about two-thirds to nearly 300 ha and included the entire area of the 2011 landslide where coherence was maintained (Fig. 8C). The maximum inferred downslope displacement rates following the landslide (14 cm/a, 9 cm/a LOS; Fig. 8D) were similar to those before it, but occurred in a region of the toe that was previously stable ('iii') and along the new headscarp ('vi') where creep had previously been slow (0.5 cm/a LOS). Displacement rates also increased in the northern part of the toe ('ii', from 2-5 to 3-6 cm/a LOS) and were more spatially variable than in the pre-failure period (Fig. 8).

Detection of movement is not possible over much of the middle and upper parts of the 2011 landslide due to decorrelation resulting from earthworks that continued for years after the event. However, because this zone is bordered on all sides by moving terrain (>1 cm/a LOS, Fig. 8C), much of it also was likely creeping throughout the period of the post-failure stack. The rate and extent of creep increased in a zone measuring nearly 500 m by 500 m (including 'v') in the paleolandslide deposit between the 2009 and 2011 landslides. Given the limited transport, and thus bulking, across most of the area of the 2011 landslide (Fig. 2A) as well as the occurrence of abundant post-failure surface displacement beyond its limits (Fig. 2B), ground motions following the event should largely represent mass movements as opposed to soil settlement or compaction.

The most striking change in ground motion following the 2011 failure is the development of a new area of creep several hundred metres wide fringing the head region of the landslide (Fig. 8C). This zone extends up to nearly 400 m beyond the headscarp on Pampahasi Plateau, in one place reaching the opposite margin of the plateau surface. The magnitude of creep decreases steadily away from the headscarp ('vi'-'x' in Figs. 8C, D, and 9B). This zone of new deformation extends hundreds of metres down both lateral margins.

All portions within the 2011 landslide area where coherence was maintained in both InSAR stacks and most areas within several hundred metres of the landslide experienced post-failure changes in slow ground motion (Fig. 10). The changes largely involved increased motion away from the satellite of between 2 cm/a and at least 5 cm/a LOS. Only a few small areas (totaling < 4 ha) in the northeast part of the toe, which were creeping at 4 cm/a LOS or more prior to February 2011, slowed following failure.

# 6 Conceptual failure model

Combining details of the 2011 failure event with patterns of long-term, pre-failure and post-failure creep, we identify several types of instability that inform the geomechanical evolution of the Pampahasi slope.

## 6.1 Components of failure

The failing slope has many interconnected components that are largely within, but also extend locally beyond, the Pampahasi paleolandslide (Fig. 11A). Slope morphology before and after the 2011 landslide shows that long-term failure occurs predominantly by complex rotational-translational sliding (Fig. 11B). Differing displacement behaviour of several zones within the slope indicate separate, although likely interacting, kinematic components. The northern part of the Pampahasi slope toe consists of an apparently competent mass that creeped uniformly at rates of 2-5 cm/a LOS prior to late February

2011 (Fig. 8B) and moved more-or-less coherently during the 2011 landslide (Fig. 2B). This block-like mass accelerated slightly to 3-6 cm/a LOS following the 2011 failure and displayed a more spatially variable pattern of deformation (Fig. 8D) suggesting partial break-up during the failure event. The resulting removal of toe support – rapidly during the 2011 landslide, but otherwise gradually – likely drove creep of the 12-ha zone of paleolandslide material ('v') directly upslope (to the northwest). Although not part of a historic landslide event (Fig. 2), this zone was creeping slowly (~0.5 cm/a LOS, Fig. 8A)

prior to the 2011 failure and accelerated thereafter (>1 cm/a LOS, Fig. 8C).

A 1-ha zone near the midline of the landslide toe ('ii') is one of the few locations of decreased post-failure activity (Fig. 10). Its high, localized activity prior to the 2011 landslide (5-8 cm/a LOS, Fig. 8C) suggests a shallow failure driven by incision at a location where Río Irpavi impinges on the toe of the slope (Fig. 2B). This localized activity is probably similar to the east-bank landslide mapped by Anzoleaga et al. (1977) ~ 400 m farther upstream, opposite the Río Jankopampa confluence

(Fig. 8A), which was creeping throughout the entire period of RADAR monitoring (Fig. 8). Decreased post-failure activity of the small west-bank instability may reflect local modification of the Río Irpavi channel by the 2011 landslide and subsequent channel remediation efforts, or it may result from post-failure stress reorganization. Several similar, linear areas of decreased post-failure creep along the toe of the 2011 landslide as far north as Río Jankopampa (Fig. 10) may represent similar conditions.

The southern part of the toe of the paleolandslide ('iii'), which was stationary prior to the 2011 Pampahasi landslide (Fig. 8A), reactivated following the failure (Fig. 8C). Although the exact cause of the reactivation is uncertain, it is clearly a result of forces imposed on the toe of the paleolandslide by the 2011 event and may be related to consequent adjustment of stream erosion along the east margin of the deposit. Creep in this area was greatest directly upslope of the place where several buildings collapsed (Fig. 6F) in the final days (February 28 and March 1) of the landslide event, suggesting that delayed

infrastructure damage there relates to the transition from the failure event to the new post-failure instability regime.

The height and steepness of the 2011 landslide headscarp indicates a deep-seated basal failure zone in the upper slope (Fig. 11B). Neither the paleolandslide nor the 2011 failure align with documented faults (Dobrovlny, 1962; Lavenu, 1978; Fig. 1B) that might indicate a structural control. A lack of displacement records for the head region following the 2011 landslide limit inferences about post-failure behaviour. However pre-failure records, although incomplete, provide some insight into

localized activity. Displacements within the head region prior the landslide and the lack of movement in adjacent steep paleolandslide scarps suggest that pre-failure creep was part of the ongoing slope failure, rather than localized instability resulting only from steep slopes. Localized, small-scale slumping occurred lower in the head region along Río Chujilluncani ('a' in Fig. 8A, B) adjacent to, or potentially within, a region of pre-failure creep (~1 cm/a LOS), supporting eyewitness accounts suggesting that the 2011 landslide initiated there.

The new displacements behind the headscarp and along the upper lateral margins of the 2011 landslide (Fig. 8B) gradually expanded over time into, respectively, the Pampahasi gravel (Fig. 9B) and paleolandslide deposits ('iv' and 'v' in Fig. 9A). Movement in the latter suggests post-failure reactivation of landslide material as a result of the 2011 landslide. Movements on the plateau are likely the result of dilation of the Pampahasi gravel in response to removal of material to a depth of up to 80 m along the 2011 headscarp (Fig. 6A).

**6.2 Mechanisms of failure**

Progressive failure is increasingly being recognized as a precursor of large, rapid landslides (e.g. Petley et al., 2002, 2005; Kilburn and Petley, 2003; Intrieri et al., 2018). In first-time brittle failures, creep reflects damage accumulation during the reduction of intact material strength, discontinuity strength, or both (Brideau and Roberts, 2015, and references therein) and consequent stress concentration (Cornelius and Scott, 1993). In such slopes, acceleration increases over time (Petley et al.,

2002; Kilburn and Petley, 2003). Eventual development of a continuous failure surface (Petley et al., 2005) enables catastrophic release, commonly resulting in debris fragmentation, high runout, and velocities exceeding 5 m/s (Hermanns and Longva, 2012). Sudden exceedance of resisting forces – for instance by seismic loading, increased pore pressure, or slope undercutting – may prematurely complete the failure surface, rapidly terminating the displacement trend (Hermanns and Longva, 2012). In slopes failing along a ductile shear zone or an existing plane of weakness, creep acceleration

decreases over time, trending toward zero (Petley et al., 2002). Because displacement typically accelerates in each of the aforementioned failure types (Fig. 11C inset), it provides an opportunity for failure forecasting (Voight, 1989; Crosta and Agliardi, 2003; Eberhardt, 2012; Baron and Supper, 2013; Hermanns et al., 2013a; Federico et al., 2015; Moretto et al., 2017).

In contrast, over the 40-month period we monitored the Pampahasi slope (Fig. 11C), long-term steady-state creep was
punctuated by temporary acceleration. Pre-2011 displacement rates are for the most part extremely slow (after Cruden and

Varnes, 1996), but constant. No progressive acceleration was evident at the temporal resolution of RADARSAT-2 or from eyewitness accounts during the days leading up to failure. Displacement rates during the four-day landslide event ranged from moderate to very rapid, but did not reach the extremely rapid range (>5 m/s: Cruden and Varnes, 1996) typical of catastrophic failures. Creep rates following the 2011 landslide were also constant, but were greater than those prior to failure, indicating a new state of instability following the landslide, which may reflect temporary post-failure adjustment. Constant creep rates suggest that the slope had, prior to failure and quickly afterward, reached steady-state behaviour, consistent with the presence of an existing failure surface (cf. Petley et al., 2002). Consequentially, prediction of large-scale failure events on the Pampahasi slope or at similar sites in La Paz should not rely exclusively on long-term creep time series.

### 6.3 Controlling factors

Geomechanical behaviour of the Pampahasi slope indicates failure nearer to that of sediment than to rock. Both the La Paz Formation and the Pampahasi gravel have compressive strengths well below 1 MPa, the threshold typically used to differentiate between rock and sediment (Brideau and Roberts, 2015). Parts of the main failure zone, as well as minor rupture zones within the landslide body, likely follow weak, fine-grained seams sheared during the Pampahasi paleolandslide or other past failures. Residual shear strength will be especially low in clay-rich zones. The lower half of the valley slope and the inferred failure zone beneath it dip at a similar angle (~8°, Fig. 11B) to the residual shear strength of the weakest (clay) zones within the failed sequence (Table S2), further suggesting that failure occurred along an existing failure surface.

Geotechnical properties of the slope, particularly its reduced strength resulting from extensive previous failure, enable its continual activity. However, external factors contribute to the long-term instability of this slope and many others in the La Paz basin. On-going fluvial down-cutting and toe erosion by Río Irpavi helps to maintain the slope's meta-stable state; spatiotemporal changes in fluvial erosion may have contributed to the differing pre-failure activity in the northern and southern parts of the landslide, as well as the increase in post-failure activity in both areas. This driver is a response to the 4-km drop in base level since the Early Pleistocene – from the time when the Altiplano plateau was internally draining to the present drainage to the Atlantic Ocean (Roberts et al., 2018) – and operates along other trunk streams in the basin.

Fewer than half of the largest historic landslides in the La Paz area follow intense precipitation events (Supplement, Table S1), suggesting that enhanced pore-water pressure may play a role in initiating some large failures. The sudden onset of the 2011 Pampahasi landslide (Fig. 11C) is typical of triggered landslides (Fig. 11C inset; Hermanns and Longva, 2012). Heavy precipitation on the previous day – the tenth wettest on record – is the only apparent trigger for the failure. Slope wetting was likely enhanced by effluent from homes and concentrated overland flow from impervious surfaces. We thus postulate that rainwater infiltrating the slope and plateau above it on 25 February reached the basal failure zone by the following evening; a consequent rapid pore-pressure increase and shear-strength reduction appear to have triggered the landslide. Rupture of the

water line crossing the slope added much more water and likely contributed to the mobility of the 2011 landslide directly downslope, where displacement rates were greatest.

## 6.4 Sources of uncertainty

Possible acceleration in the days or weeks prior to the 2011 landslide cannot be evaluated with the temporal resolution of the displacement histories used here. RADARSAT-2's 24-day revisit period provides low sampling frequency compared to some other RADAR satellites (Moretto et al., 2017) and especially to in-situ sensors or ground-based remote sensors (e.g. Kalaugher et al., 2000; Crosta and Agliardi, 2003; Gischig et al., 2011; Eberhardt, 2012; Federico et al., 2015; Confuorto et al., 2017; Carlà et al., 2018). Additionally, the temporal filter used in processing (Supplement, section 4.6) attenuates records of non-linear displacement trends. The resulting temporal details reported here are sufficient to characterize gradual acceleration over periods of months (Fig. 9), as has been documented for pre-failure creep using InSAR elsewhere (Carlà et al., 2018; Intrieri et al., 2018), or more abrupt acceleration over just a few RADAR acquisitions, but not shorter term changes. Sporadic displacement activity that may signal impending acceleration (cf. Kalaugher et al., 2000) could similarly have gone undocumented.

Ground motion represented in displacement maps is independent of the structural behaviour of the built environment. Isolated building instability is likely in light of some local construction practices in La Paz, but will be extremely localized and thus is removed during spatial interpolation of the maps. Phase change due to thermal expansion will be minimal given limited seasonal temperature differences in the study area. Due to their cyclic nature, any such phase component will not influence long-term displacement trends.

The HDS-InSAR processing chain is complex and includes many non-linear steps, which greatly complicates development of an accurate error-approximation model. Both the pre-failure and post-failure stacks have good baseline diversity, allowing relative errors between them to be approximated by first order estimates from the square root of the number of scenes (32 vs. 12). In the absence of a rigorous model, we assume that error for the thinner stack is conservatively twice as large as that of the thicker stack. We approximate the errors as 3 mm/a and 6 mm/a, respectively, for the pre-failure and post-failure stacks. Because the average displacement rates across much of the Pampahasi area exceed the error estimates, the displacement patterns documented here are deemed to be reliable.

Due to the structure of the HDS-InSAR processing chain, differing environmental conditions between the two stacks, namely topography and moisture (cf. Wasowski and Bovenga, 2014, 2015), have minimal effects. The reference digital terrain model (DTM) is used only for an initial topographic correction; stack processing solves for height error relative to the reference DTM and provides a new elevation solution for each of the two stacks (Supplement), which improves terrain representation. Topographic correction of the post-failure stack thus accounts for landslide-induced terrain changes, which

were greatest in the source area (Fig. 11B). Temporal soil moisture variability is unlikely to affect phase by more than 100° (Rabus et al., 2010), which equates to approximately one-third of an interferometric fringe or 0.9 cm for the sensor used here. Comparison of precipitation records in the 30 months before and 10 months after the 2011 Pampahasi landslide indicates that long-term precipitation amounts during the pre-failure and post-failure stacks were comparable (Fig. S2; Table S3). Spatial moisture gradients are a more substantial error source (Rabus et al., 2010), but major differences present in a single scene are removed during stack processing.

The assumption of slope-parallel motion is likely incorrect in some parts of the Pampahasi area. The complex nature of the 2011 failure, particularly the variability in the displacement directions (Fig. 2B), suggests that separate masses may have moved obliquely relative to one another during pre-failure and post-failure creep. Localized compression or extension within the landslide would additionally impart vertical components of movement. Additional error is thus introduced in the conversion from measured one-dimensional (LOS) displacement to approximate three-dimensional (true) motion. Provided that the sense of movement at any given location on the slope has not substantially changed following the 2011 failure, LOS motion vectors record similarly scaled changes in the true rate of motion and thus reliably represent the degree of change in slope activity. Additionally, the close alignment between the RADAR LOS and the fall-line of the slope ensures that the InSAR-measured displacement record is particularly sensitive to expected gravity-driven slope motion between Pampahasi Plateau and Río Irpavi.

Because RADARSAT-2 acquisitions over the site ended 10 months after the failure, the current activity of the slope is poorly constrained and it is possible that increased post-failure slope creep was short-lived. Additional InSAR acquisitions planned for this and other sites in La Paz, including both ascending and descending orbits, will provide additional insight on more recent temporal and special complexity slope activity.

## 7 Advances in InSAR monitoring of landslides

### 7.1 Comparison with other studies

Our detailed documentation of slope activity preceding and following a large landslide using spaceborne InSAR is, to our knowledge, unique. Numerous studies show evidence of long-term, generally sustained creep of slow-moving landslides (e.g. Colesanti and Wasowski, 2006; Lauknes et al., 2010; Hermanns et al., 2013b; Necsoiu et al., 2014; Schlögel et al., 2016), and in many cases document temporal variability in displacements. A much smaller number of studies identify short-term deformation between successive scenes (Moro et al., 2011) or creep leading up to catastrophic failure (Carlà et al., 2018; Intrieri et al., 2018; Handwerger et al., 2019). However, we are aware of only one other spaceborne InSAR investigation that compares pre-failure and post-failure activity. Confuorto et al. (2017) report displacement rates during a

one-year period starting ~20 months following the 2012 Via Piave landslide in southern Italy that were marginally faster than rates during a ca. two-year period ending ~16 months before failure (See Supplement, section 5, for comparison with the dataset presented here). In that investigation, the limited number of point reflectors, their concentration in a narrow band of the headscarp, and the three-year data gap bracketing the landslide limit insight into potential post-failure changes of the slope.

## 7.2 Expanding utility

Advanced InSAR approaches such as HDS-InSAR increase ground-target density and discriminate spatially uncorrelated deformation, improving the characterization of landslides with spaceborne InSAR. Even though applied here to RADAR data of moderate spatial and temporal resolution, HDS-InSAR has provided detailed characterization of a large, dynamic urban slope. Applying the technique to datasets with higher spatial resolution will allow even more detailed characterization of landslide creep, and sensor constellations will help increase the temporal resolution of displacement records. The HDS technique also provides insight into the precise locations of landslide margins and is thus particularly useful in dense urban settings where development and remedial works remove evidence that is useful in landslide investigations.

Furthermore, such techniques enable more comprehensive detection and characterization of landslides of different depth and size. Preferential detection of deep landslides by InSAR (Wasowski and Bovenga, 2014, 2015) reflects the typically high spatial regularity of their displacements. Shallow instability, especially in areas of variable micro-topography, generates spatially irregular ground motion that is more difficult to detect. HDS-InSAR's optimization for locally variable ground motion particularly improves the characterization of shallow landslides and thus reduces biasing toward deep-seated instability.

## 8 Conclusions

The 2011 Pampahasi landslide is only one event in the long-term, and likely on-going, activity of a meta-stable slope. High-spatial-density InSAR results demonstrate that several parts of the slope were active over the years prior to the landslide and that nearly all regions with coherent ground-motion records within and adjacent to it experienced post-failure increases in the extent and rate of creep. This change in ground deformation counters any expectation that a complex landslide might stabilize, at least temporarily, following a discrete failure and highlights that such an assumption, at least for short-term stability, is imprudent for multi-generational failures. Slope dynamics documented here support observations and theory from the scientific literature that spatiotemporal clustering of landslides are responses to stress redistribution, which is likely to exceed stress changes due to background erosion.

InSAR-derived deformation patterns provide new insight into failure mechanisms of the Pampahasi slope. The complex failure comprises multiple unstable parts, including many areas of shallow instability and at least one large creeping block. Accelerating creep, a typical feature of first-time and reactivated failures, was not apparent leading up to the disastrous landslide on 26 February 2011. Any increase in displacement during the 13-day window following the last pre-failure

RADAR scene (13 February) apparently was not perceived by residents on the slope. The absence of detectable acceleration may instead reflect the achievement of steady-state creep along an existing surface, which is supported by the presence of previously failed sediments and the similarity of the gradient of lower slopes to the residual strength of the weakest geologic units. Increased pore pressures resulting from particularly heavy rainfall triggered a large failure that temporarily perturbed the dynamic equilibrium of the slope. The failure reorganized stresses in the slope as well as beneath the plateau above it,

leading to a new, although possibly temporary, state of dynamic equilibrium affecting even more of the slope.

Improved understanding of instability of the Pampahasi slope is instructive in evaluating and reducing risk from landslides in La Paz, especially given the large extent of paleolandslides in the city, their close association with recent landslides, and incomplete knowledge of slope activity. The Pampahasi slope is similar to numerous other slopes formed on large ancient landslides in the city, including slopes where many of the largest historic landslides have occurred (Fig. 2B). These slopes

are underlain by generally weak, fine-grained sediments of the La Paz Formation. Such slopes, particularly those descending from the west margin of Pampahasi Plateau, are key sites for future stability and risk assessments. Additionally, stress redistribution suggested by accelerated creep of the Pampahasi paleolandslide between the margins of the 2009 and 2011 failures and possibly in adjacent parts of the Villa Salomé paleolandslide may increase their susceptibility to large-scale rapid failure in the future. Evaluating the possibility of future, large-scale reactivations of complex landslides should not,

however, rely solely on creep acceleration, especially given the lack of such a signal leading up to the 2011 landslide. Furthermore, although fewer than half of the historic failures exceeding 1 $Mm^3$ happened during the rainy season, coincidence of the 2011 reactivation of the Pampahasi paleolandslide with particularly wet conditions indicates that consideration of high-rainfall scenarios is advisable. In light of the evidence presented here of enhanced post-failure activity, creeping slopes in La Paz are generally not appropriate for new settlements or resettlement of residents displaced by

disasters.

Optimizing InSAR characterisation of landslides requires processing methodologies that afford high ground-target density and discriminate spatially uncorrelated deformation. The improved understanding of the activity of the Pampahasi area reflects the detailed spatial characterization made possible by the HDS-InSAR technique. Applying advanced InSAR processing techniques such as this to modern and next-generation RADAR datasets that, due to continued technological

improvement, have higher ground resolution and revisit frequency will improve understanding of complex unstable slopes.

## Author contribution

NJR, JJC, RLH, and BTR conceived the investigation. NJR processed RADAR datasets with guidance from BTR. All authors participated in fieldwork. MAG and EM arranged field logistics and provided crucial details on the history of development and instability. NJR, JJC, BTR, and RLH interpreted observations and results. NJR prepared the manuscript with contributions from all co-authors.

## Competing interests

The authors declare that they have no conflict of interest.

## Acknowledgements

NJR used facilities and software at MacDonald, Dettwiler and Associates (MDA) courtesy of H. Zwick and C. Nadeau, and was assisted in-house by J. Sharma and J. Eppler. He was partially funded by the Natural Science and Engineering Research Council (NSERC) of Canada (Postgraduate Scholarship) and the American Society for Photogrammetry and Remote Sensing (ASPRS; Robert C. Colwell Fellowship). MDA Geospatial Services provided RADARSAT-2 imagery. Field investigation was supported by NSERC (Discovery Grant 24595 to JJC), APSRS (Ta Liang Memorial Award to NJR), and the International Centre for Geohazards (ICG Contribution no. 377 to RLH). Victor Ramírez and Eddy Baldellón (Municipality of La Paz) relayed eyewitness accounts of the 2011 Pampahasi landslide reported by residents of the affected area. Detailed comments and suggestions from anonymous reviewers significantly improved the final paper.

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

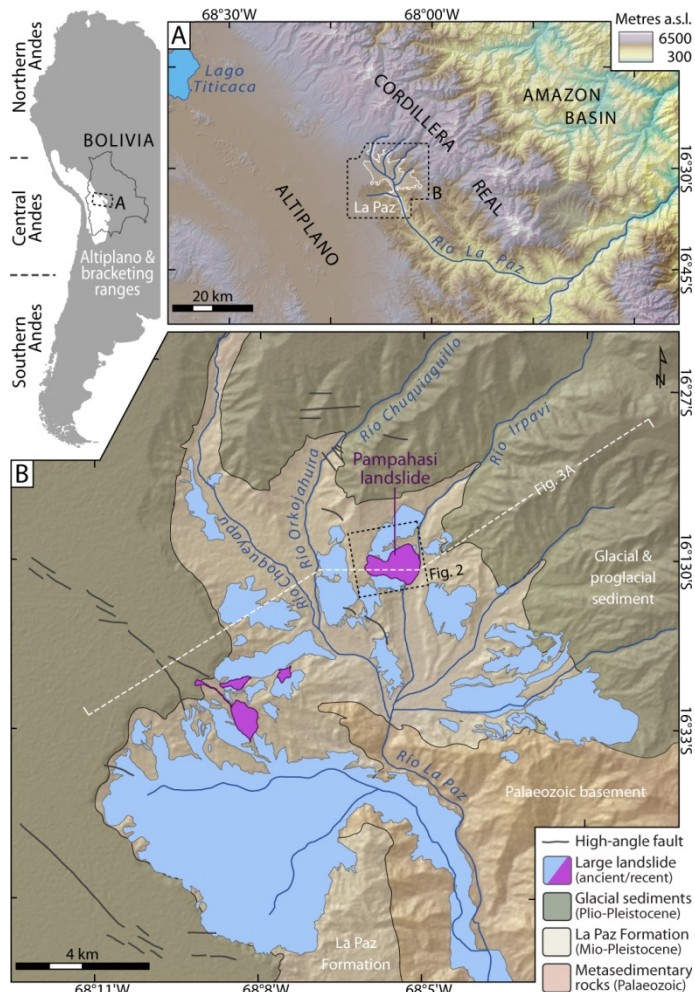

**Figure 1. Setting of La Paz, Bolivia. A.** Location and physiographic context. **B.** Generalized geology after Dobrovolny (1962) and Anzoleaga et al. (1977), with faults from Dobrovolny (1962) and Lavenu (1978). Terrain is from the ASTER GDEM 2 produced by METI and NASA.

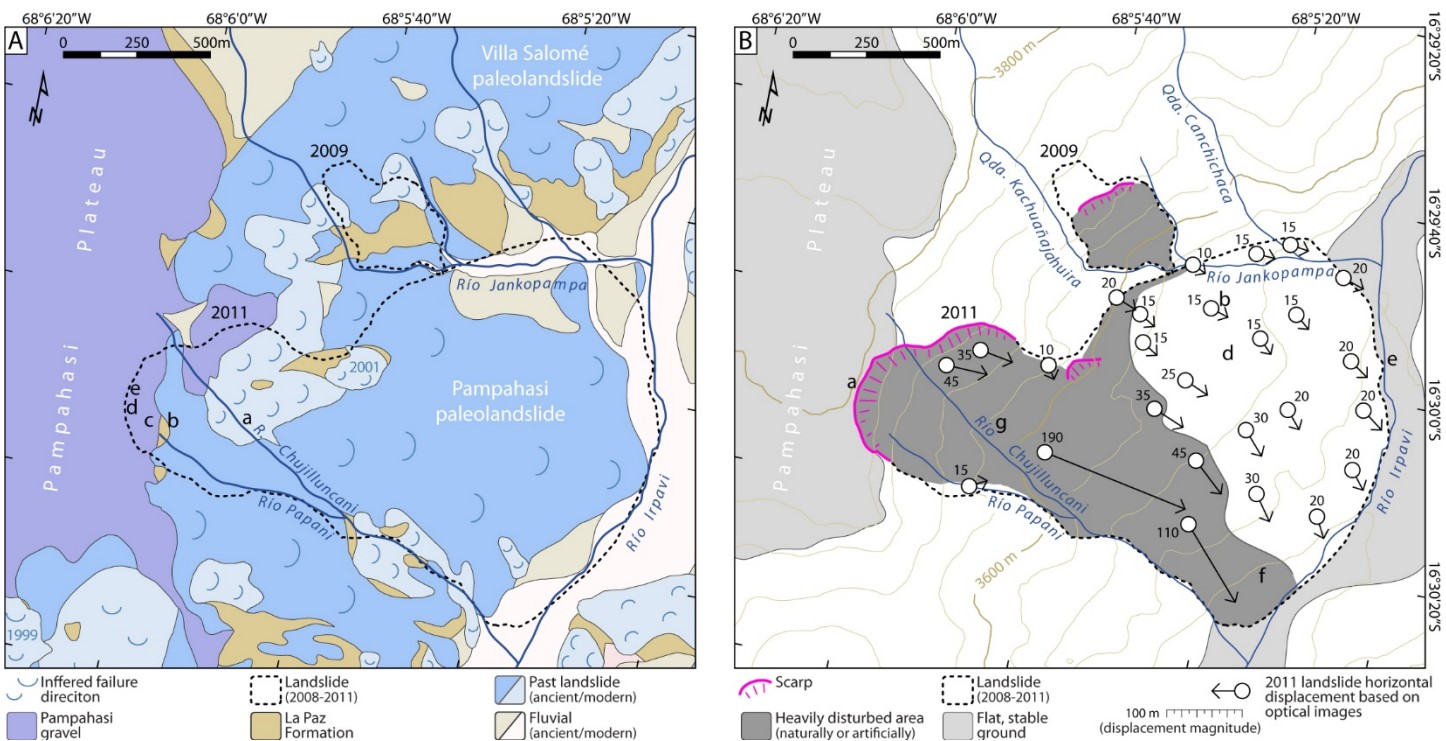

**Figure 2. Ground surface conditions in the Pampahasi area. A.** Surficial geology prior to the start of the twenty-first century; adapted from Anzoleaga et al. (1977). **B.** Slope morphology and ground disturbance showing the extent and morphologic features of landslides that occurred during the period of RADAR acquisition (September 2008 to December 2011). Horizontal displacements during the 2011 Pampahasi landslide are based on comparison of pre-failure and post-failure WorldView-2 scenes. Contours are from NASA's Shuttle RADAR Topographic Mission V3.0.

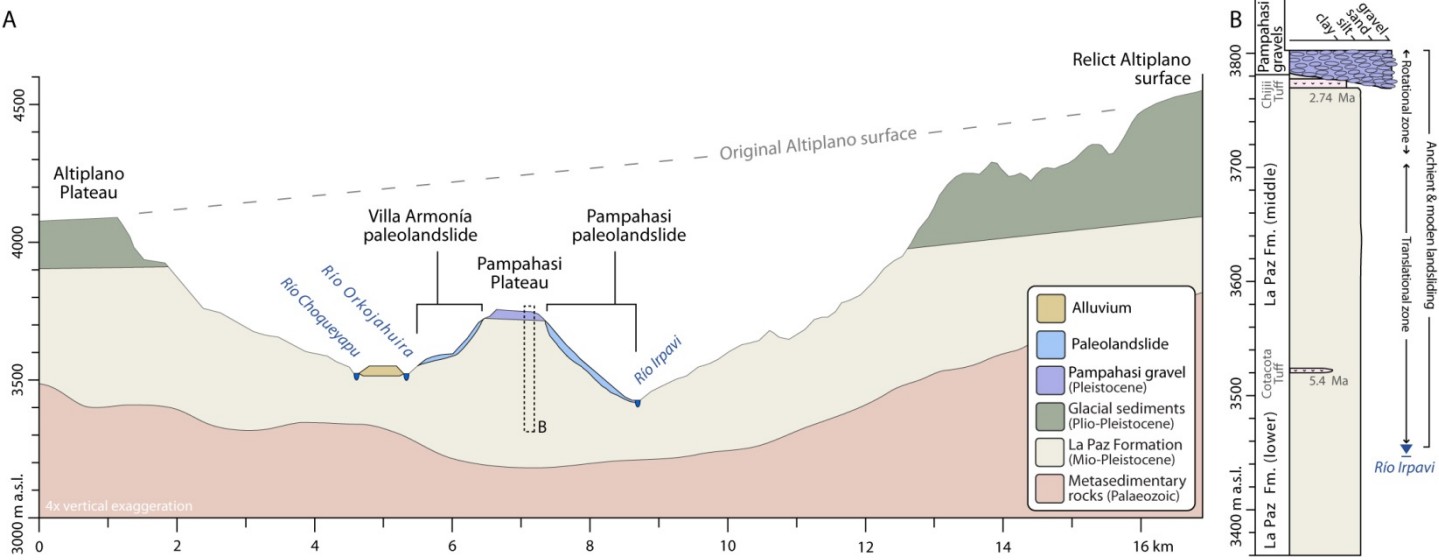

**Figure 3. Stratigraphic sequence. A.** Cross-section oblique to the trend of the Cordillera Real through the fill underlying the Altiplano (location in Fig. 1B). **B.** Lithostratigraphy beneath Pampahasi Plateau, including principle units involved in the Pamphasi paleolandslide and 2011 Pampahasi landslide. Modified from Anzoleaga et al. (1977) based on field mapping throughout the La Paz area.

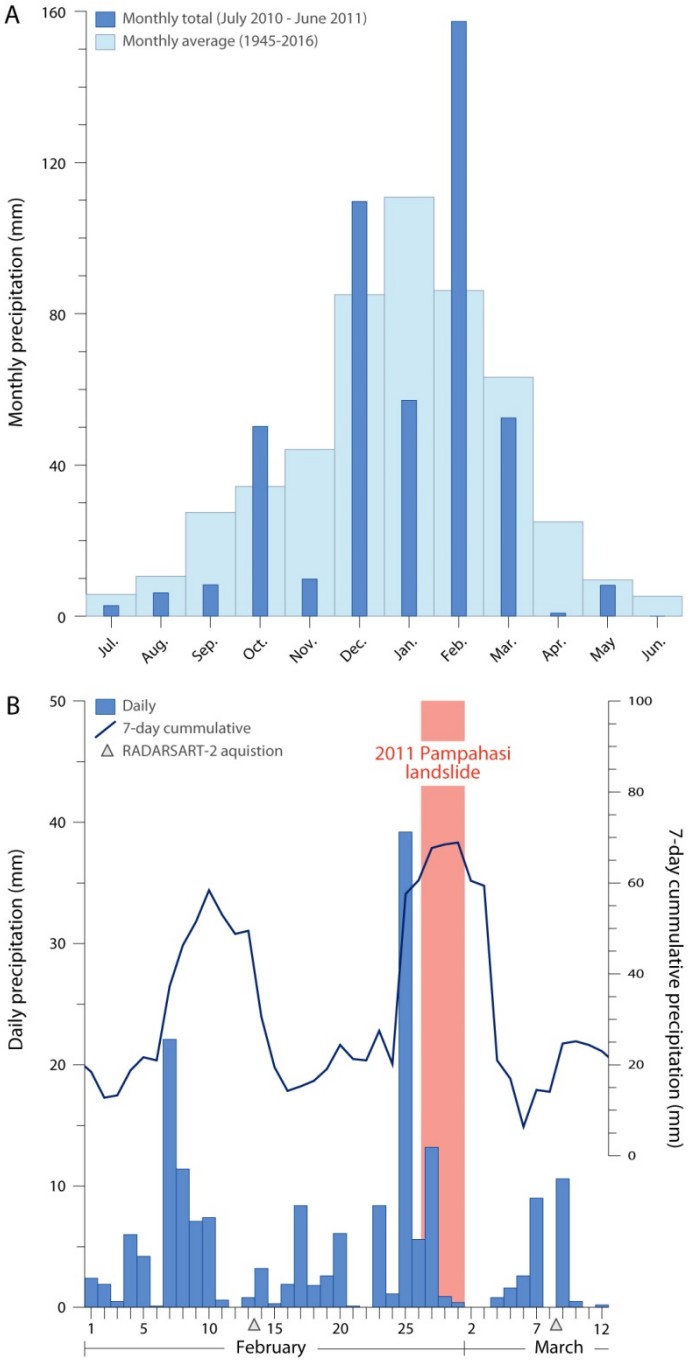

**Figure 4. The precipitation record for the Laykacota meteorological station (see Fig. S1 for station location). A.** Comparison of monthly average precipitation (1945-2016) with monthly precipitation for the second half of 2010 and first half of 2011. **B.** Daily precipitation and seven-day cumulative precipitation during February and early March 2011.

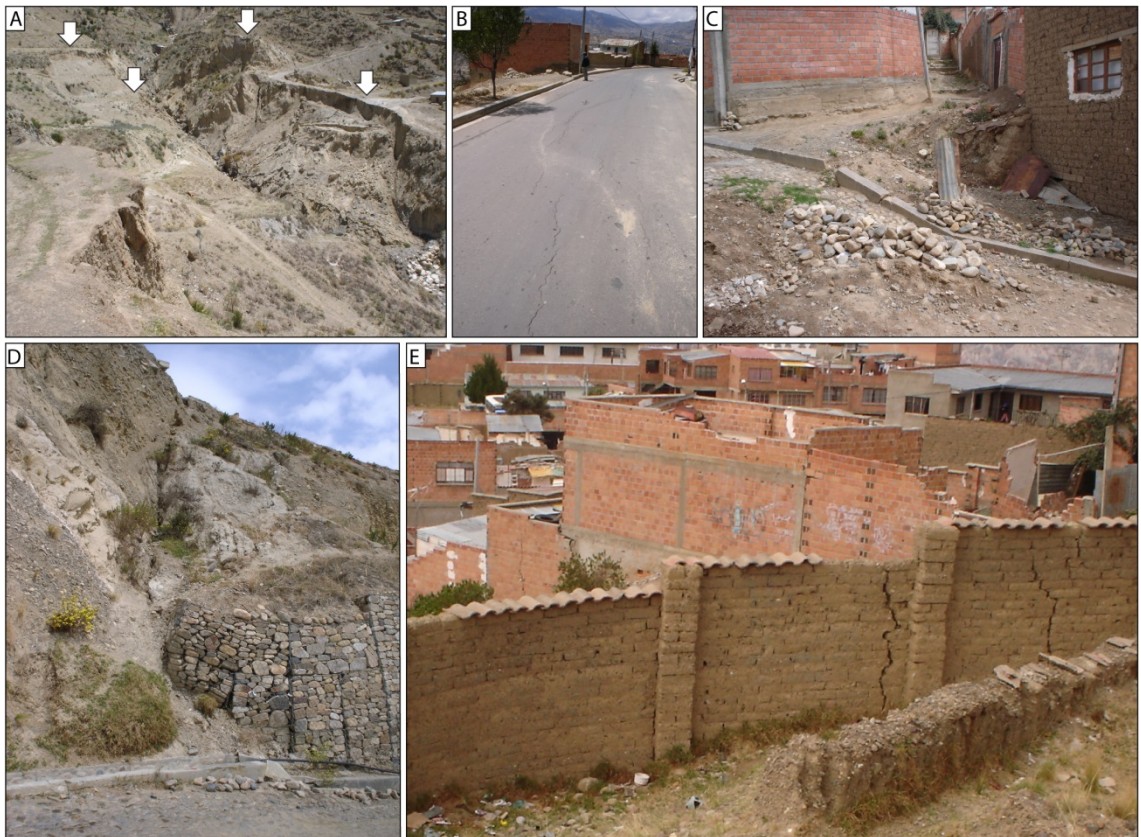

**Figure 5. Field evidence of pre-failure creep in what would become the head region of the 2011 Pampahasi landslide. A.** Small recent slumps (arrows) along Río Chujilluncani, ~400 m downslope from the east margin of Pampahasi Plateau. **B.** Minor cracking and centimeter-scale vertical offset of a paved road about 200 m east of, and 75 m below, the east margin of the plateau. **C.** Unpaved road with ~1 m displacement near the 2011 headscarp. **D.** Vertical offset along a gully ~75 m from the future south lateral margin of the 2011 landslide. The side of the gully nearer the middle of the 2011 landslide (right of the photo) has dropped ~0.5 m relative to the adjacent slope, causing deformation of the gabions and curb in the foreground. **E.** Cracks in an adobe wall (right foreground) and brick house (left middle ground) ~50 m downslope of the east margin of the plateau, corresponding to the middle of the 2011 landslide headscarp. See Figures 2A and 7A, B for locations of photographs. Photos taken by R.L. Hermanns in October 2005.

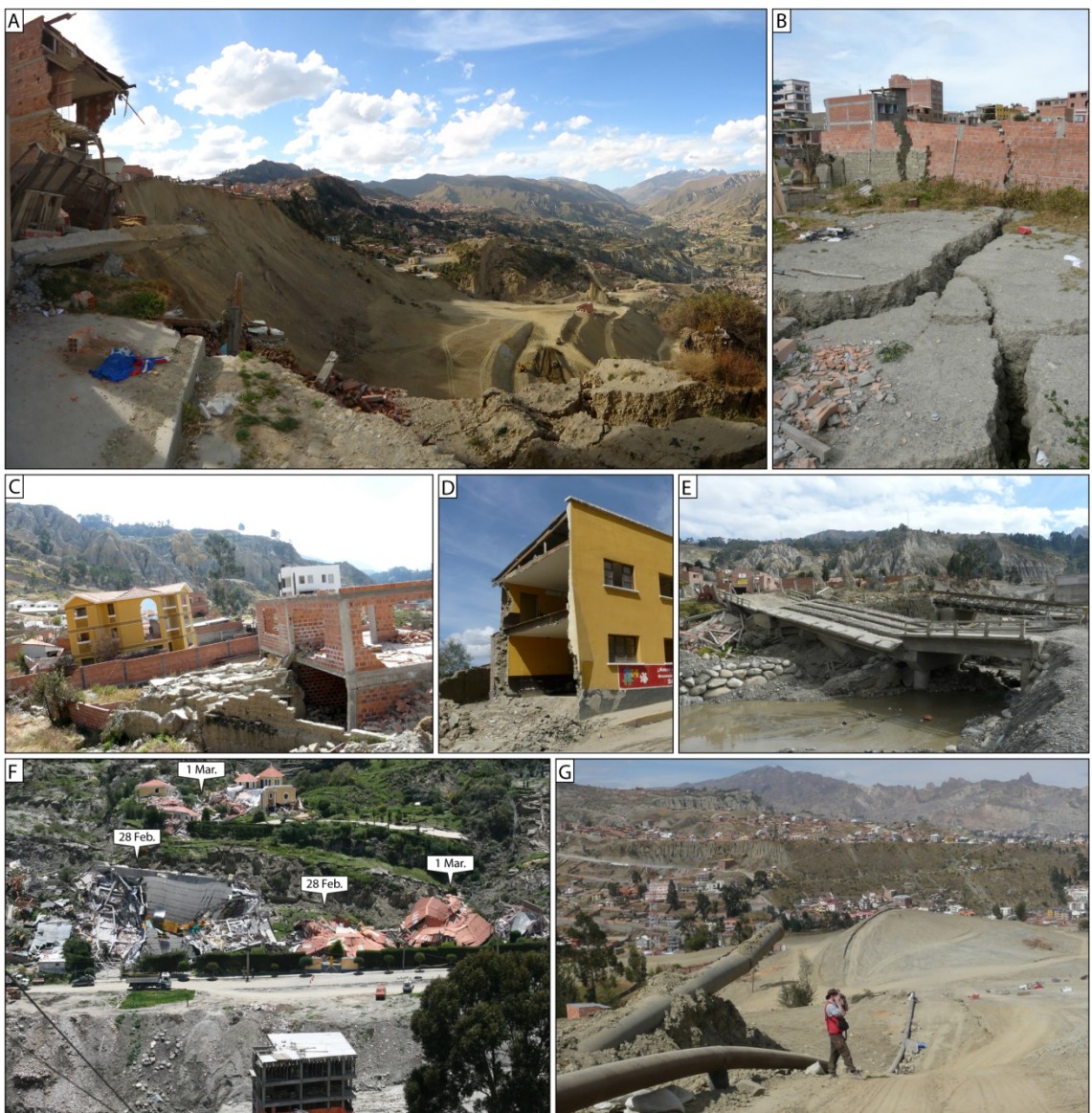

**Figure 6. Impacts of the 2011 Pampahasi landslide. A.** Headscarp (80 m height) along the east margin of Pampahasi Plateau (note excavator and trucks in lower middle ground for scale). **B.** Ground fissures in the northern part of the lower landslide mass. The wall in the background is ~2 m high. **C.** Back-tilted houses and apartment buildings on the northern part of the lower landslide mass (movement was from left to right). **D.** Community building with collapsed wall on the northern part of the lower landslide mass. **E.** Bridge across Río Irpavi that was deformed and shifted from its footings on 28 February near the centre of the landslide toe. **F.** Buildings on the southern part of the toe of the landslide that collapsed on 28 February and 1 March (dates of collapse shown; photo by M.A. Guzmán, 2 March 2011). **G.** Water service line ruptured by the landslide (upper pipe) and replacement water service line (lower pipe). See Figures 2B and 7C, D for locations of photographs. Photos taken by N.J. Roberts in May 2011 unless otherwise indicated.

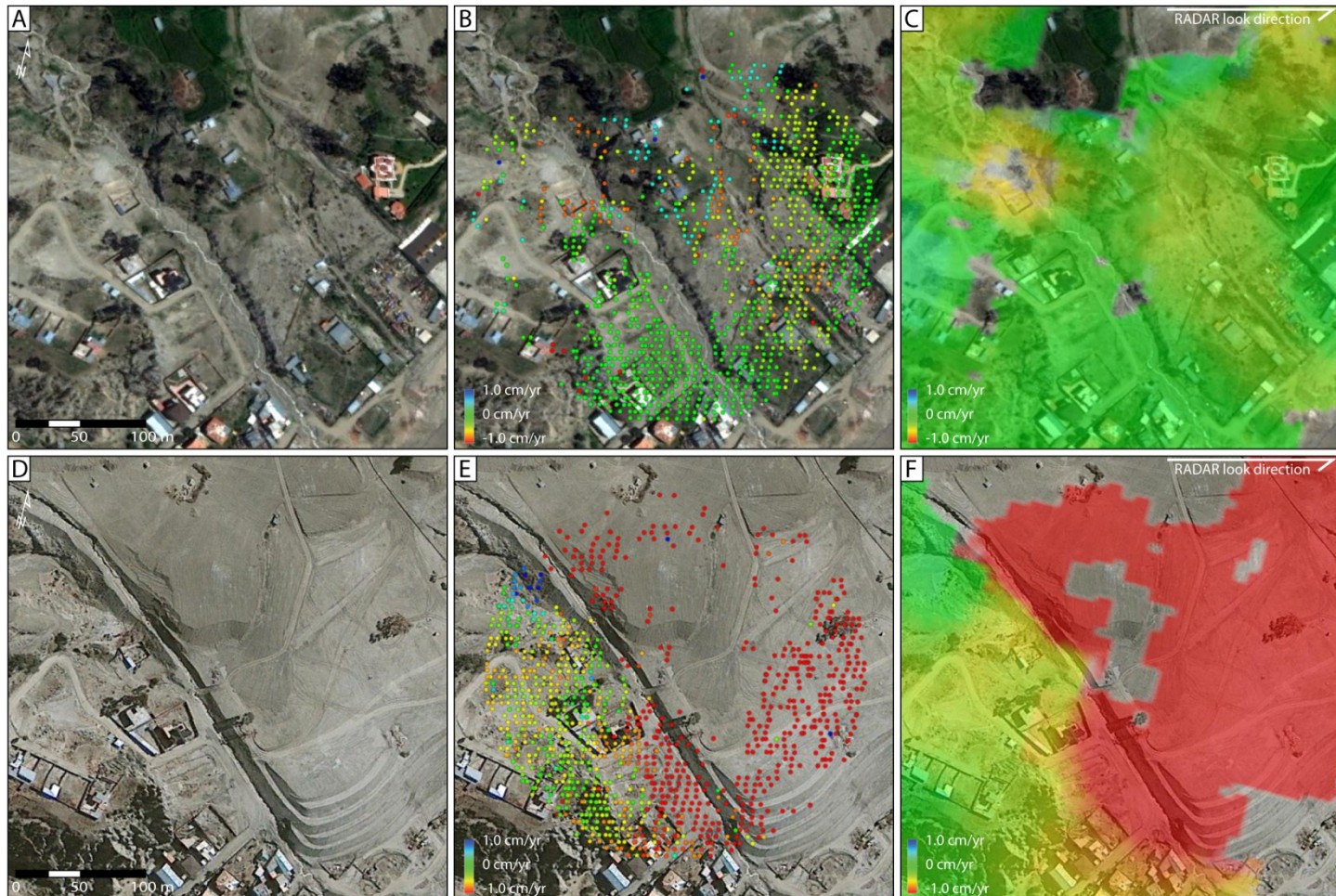

**Figure 7. Typical density and spacing of HDS-InSAR results for an area of mixed suburban, agricultural, and commercial development.** Both the pre-failure stack (**A**, **B**, **C**) and post-failure stack (**D**, **E**, **F**) are shown**.** Panels For each stack depict: **s**atellite image showing land cover without InSAR data (**A**, **D**); point data representing HDS neighbourhoods (pixel neighbourhoods) (**B**, **E**), each recording a displacement time history; and linear displacement map interpolated from HDS points (**C**, **F**). Localized variability in HDS results is suppressed by linear deformation interpolation. Displacement rates are in the satellite line of sight. The deforming area is the southern boundary of the active part of the landslide toe (see Fig. 7 for location). Base image is a 4 January 2011 Quickbird image viewed in GoogleEarth™. Base images are from GoogleEarth ([A, B, C] 4 January 2011 and [D, E, F] 25 July 2011).See Fig. 8A, C for location.

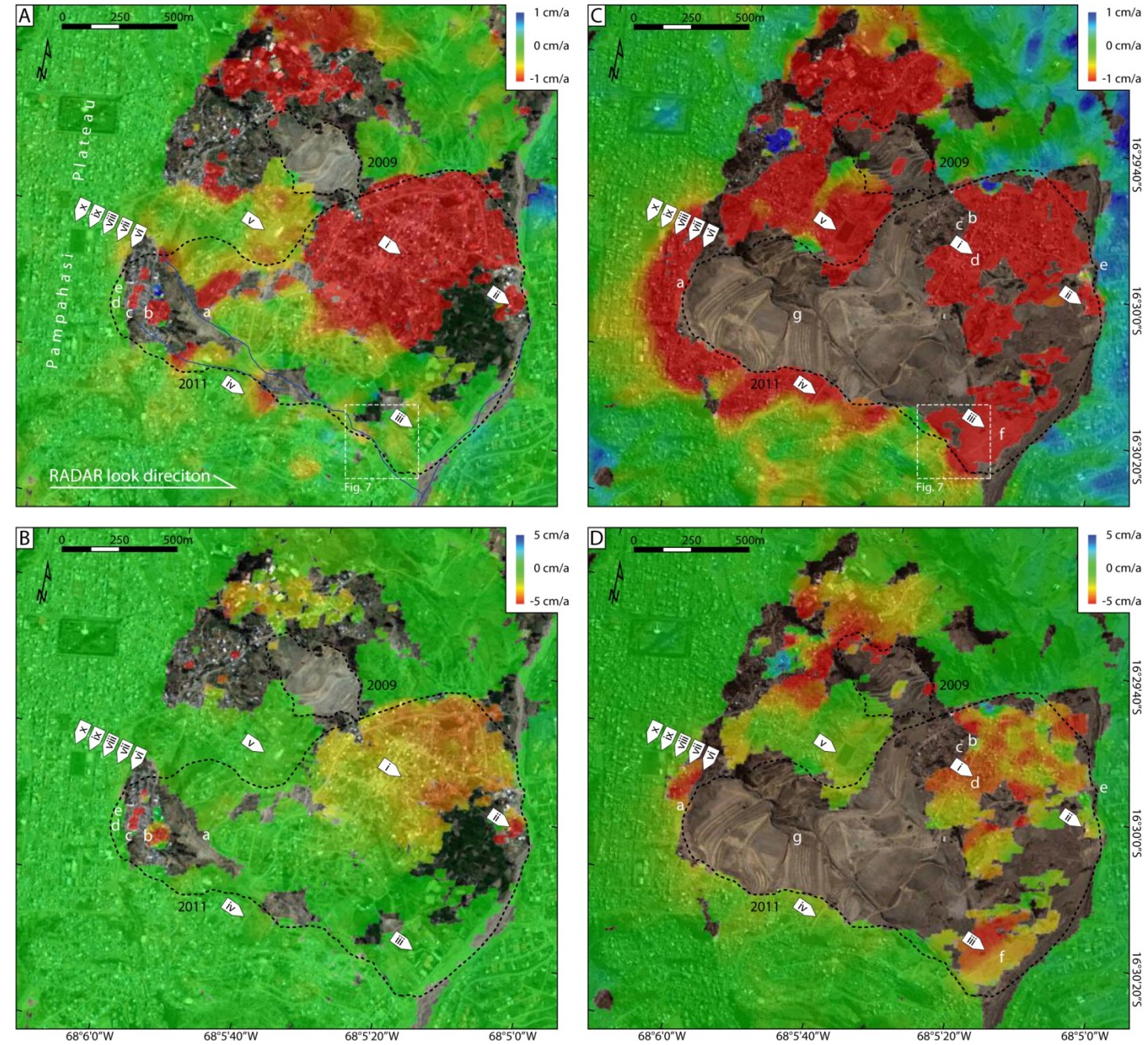

**Figure 8. InSAR-measured line-of-sight ground displacements in the Pampahasi area.** Average linear deformation rates during (**A**, **B**) the 30-month period before and (**C**, **D**) the 10-month period after the Pampahasi landslide. Compressed (-1 to 1 cm/a; A and C) and expanded (-5 to 5 cm/a; B and D) displacement scales emphasize, respectively, the spatial limits and spatial variability of slope activity. Dashed lines are the limits of landslides that occurred in 2009 and 2011 during the period of RADAR scene acquisition. Roman numerals

show locations of displacement histories in Figure 5. Letters show locations of photos taken, respectively, before (Fig. 5) and after (Fig. 6) the 2011 landslide. Base images are from GoogleEarth ([A, B] 4 January 2011 and [C, D] 25 July 2011).

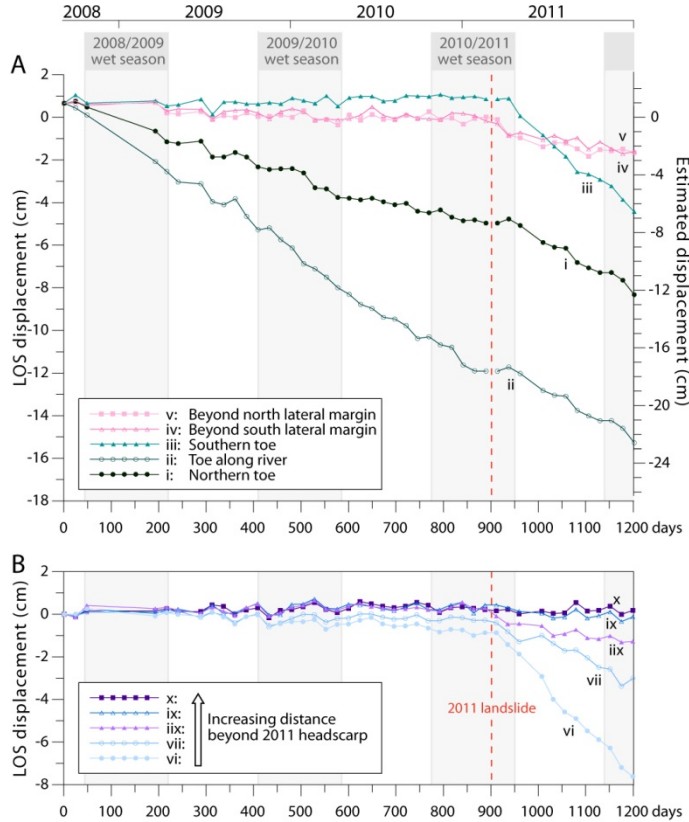

**Figure 9. Displacement time series from the Pampahasi area (locations 'i' to 'x' in Fig. 8). A.** Movement of the west slope of the Río Irpavi valley within the area affected by the 2011 landslide (i-iii) and adjacent paleolandslide deposits (iv-v). The right axis is approximate, given that additional error is introduced from conversion from LOS to true motion. **B.** Movement on Pampahasi Plateau behind the headscarp of the 2011 landslide (vi-x). Variability in the data, which is most pronounced in parts of the time series with low displacement rates, indicates phase noise resulting from factors unrelated to ground motion. Displacements between the end of the pre-failure stack and the start of the post-failure stack are not precisely constrained within the limits of the 2011 landslide (i-iii). For the Pampahasi slope adjacent to the 2011 failure and for the Pampahasi Plateau, displacements between the stacks are based on results of full-stack processing (see Supplement, section 4.8).

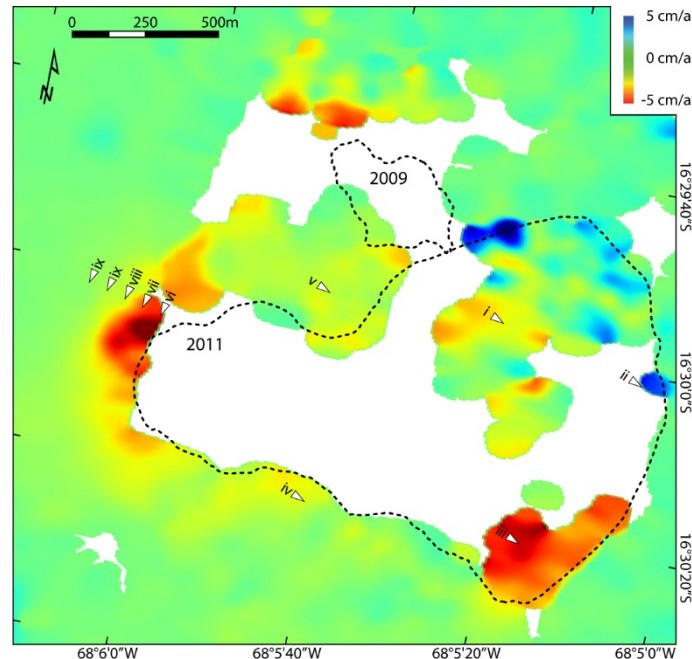

**Figure 10. Change in line-of-sight slope creep of the Pampahasi area following the 2011 failure relative to pre-failure creep.** Positive and negative values indicate, respectively, decreased and increased post-failure displacement rates away from the satellite. Green areas experienced no change in displacement and were largely stable both before and after the 2011 landslide. White areas lacked coherence phase in pre-failure scenes, post-failure scenes, or both. Roman numerals show locations of displacement histories in Figure 5. Pre-failure and post-failure imagery spans, respectively, 08 September 2008 to 13 February 2011 and 09 March 2011 to 22 December 2011.

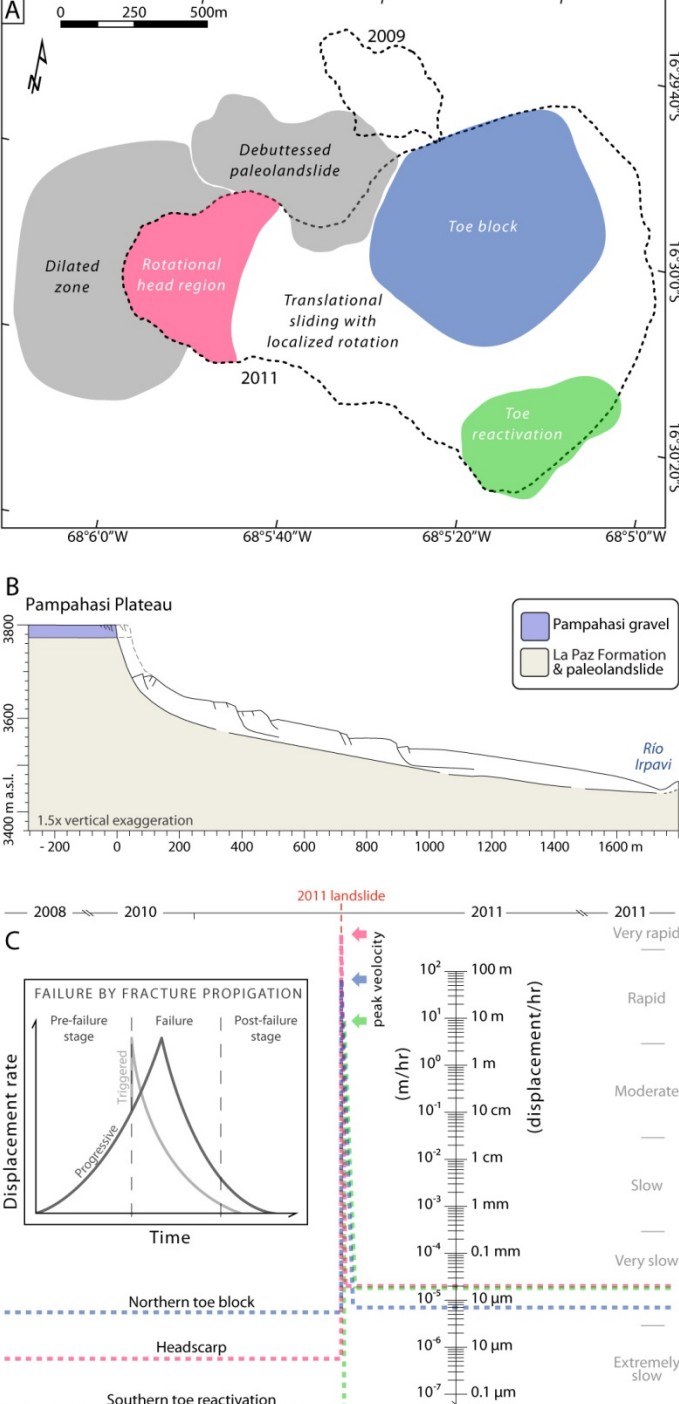

**Figure 11. Proposed geomechanical model for the Pampahasi landslide and adjacent terrain. A.** Components of failure, which differ in kinematics and activity history. **B.** Generalized section through the failing slope illustrating the approximate depth and attitude of the failure zone. The topographic surface is based on NASA's Shuttle RADAR Topographic Mission V3.0, which depicts the generalized pre-failure terrain. Dashed and solid line work in the source area (0 to ~150 m x-axis distance) represent, respectively, pre-failure topography and post-failure topography based on terrain changes measured in the field using a handheld laser rangefinder. Terrain lower on the failed slope changed comparatively little during the landslide; solid line work beyond ~150 m x-axis distance is thus generally representative of both the pre-failure and post-failure topography. **C.** Schematic temporal displacement history generalized from InSAR measurements bracketing the 2011 Pampahasi landslide and eyewitness accounts. Due to the revisit frequency of RADARSAT-2 (24 days), the lack of acceleration leading up to the 2011 Pampahasi landslide is based on the lack of such effects reported by local residents. Velocity classes are from Cruden and Varnes (1996), spanning all classes except for 'extremely rapid' (>5 m/s). Triangles denote timing of RADARSAT-2 acquisitions (dates provided in Table S4). Inset shows schematic stages of slope movement during first-time failure by fracture propagation (after Hermanns and Longva, 2012).

