# Peer review of "Changes in ground deformation prior to and following a large urban landslide in La Paz, Bolivia, revealed by advanced InSAR"

_Natural Hazards and Earth System Sciences, 2018_

## Referee Comment (RC1) · Anonymous Referee #1 · 20 Aug 2018

Dear authors,

The presented paper contains very interesting and useful approach and results for landslide movement studies with important implications for the hazard assessment applicable in regions where landslides are major problem. It is very well written and presented and the results are quite interesting for wide audience. But I think that your interpretations are not well supported by the results and available data and that you missed important geomorphological evidences with important implications for the result interpretation. I think that the studied landslide is not suited for investigations of the simple post-failure movement behavior which suggests at least temporal stabiliza-

tion of the landslide after its major failure. Plus the InSAR results need to be combine with detailed, site specific geomorphological/landslide mapping which was not done (or at least was not presented) and therefore I am convinced that some of the interpretations should be changed/improved. Your statement challenging the post-failure "stabilization" theory are not well documented in the article: You did not defined the "hypothesis" about stabilization after failure - how do you assess the "stability"? I am sure that factor of safety of the main sliding plane calculated after the event would be much higher than before, clearly showing the slope stabilized. Surface movements observed by InSAR technique do not necessarily represent failure plane movements. As far as I know, the post failure "stabilization" has never been the only or major hypothesis about landslide movement. There is very common concept of post-failure adjustment which includes increased activity mainly around the scarp and toe if river is eroding it and locally on sites with steep slopes. Moreover, the landslide you describe represents reactivation of deep and complex landslide and in such cases (regions with long term, complicated landslide history) it has been observed before that reactivation of one part may trigger activity in other areas (domino effect). Whereas I think that the theory of post-failure stabilization has always been limited rather to simple landslide cases (which are not the one you describe). Therefore I disagree with interpretation of your measurements – they very nicely document complex landslide behavior which correct (as far as I can tell from looking at the Earth Google images) explanation would require detailed geomorphological interpretation of the well morphologically defined "paleo landslide" (yellow line on the attached Fig. 1 EarthGoogle image below), which N limit is some 400 from the 2009 landslide well defined by escarpment. It seems that this paleo landslide body is strongly segmented by gullies running to the main river valley as well as significant slope forming toe of the 2009 landslide above the creeping region with "i" on Fig. 7A.

For further comments, please see the attached files. Please, consider the attached figure only as a suggestion.

Please also note the supplement to this comment:
https://www.nat-hazards-earth-syst-sci-discuss.net/nhess-2018-211/nhess-2018-211-RC1-supplement.zip
* * *
[Figure]

**Fig. 1.**

---

## Referee Comment (RC2) · Anonymous Referee #2 · 13 Sep 2018

This is a good, interesting paper. I enjoyed reading it. The authors clearly have good complementary expertise needed to show how advanced InSAR can provide very useful information on ground surface deformations affecting unstable and marginally stable slopes. The displacement data pre- and post-dating the occurrence of a well-documented large urban landslide are valuable and rather unique. The data analysis is original and shows the authors appreciate the complexity of slope/landslide deformation patterns, as well as the strength and limitations of InSAR.

Specific comments for the authors' consideration.

1) Data quality/relevance - The authors discuss some limitations of InSAR data and

mention that the quality of the post-failure InSAR results is expected to be lower than that obtained from the thinner stack of the pre-failure radar acquisitions. This is correct, but since you compare the InSAR results from two different periods, I think the issue of data quality deserves more attention. The precision of InSAR measurements depends also on the environmental conditions and the adopted coherence threshold (cf. Wasowski and Bovenga, 2014a,b). For instance, after the Feb 2011 failure, the topography of the slope has changed with respect to the reference SRTM DEM. Significant change for InSAR processing and sensitivity to displacement? Representative pre-failure and post-failure topographic sections of the landslide could help. Also, was the weather less rainy (dry) in the post-failure period with respect to the pre-failure time? Significant for the processing results? Fig 6b, which shows the distribution and average annual velocity of HDS (measurement points) for the pre-failure period, indicates some "noise" (especially southern part) in the data (areas where different velocity points are mixed together). Perhaps it would be useful to show a similar figure for the post-failure period. If not, to give an idea of the precision of the results obtained for the pre- and post-failure, you could consider estimating the mean velocity standard deviations for the two periods (cf. Wasowski and Bovenga, 2014a,b). Finally, we know the InSAR are relative in both time and space. You indicate the Master scene in the Supplement (for both pre- and post-failure stacks), but the location of a reference point (area) in space is not specified, unless I missed it. This could be relevant considering the generally marginal (changing?) stability of the land in the study area.

2) Interpretations and conclusions - I can follow you for the most part, but remain somewhat uncertain about the postulated broad significance of the observed post-failure creep acceleration (enhanced activity). One reason is that the quality/precision of pre- and post-failure measurements is not the same and perhaps difficult to assess (cf point above). Then, the conclusion regarding the post-failure creep acceleration is based just on this one specific case. On the basis of the literature review and their own data, Wasowski and Bovenga (2014a,b) indicated that InSAR seem to preferentially capture creep of deep slides, seasonal accelerations of large landslides and

post-failure ground instability (settlements, volumetric changes). For some deep land-slides/materials, these settlements and volumetric changes can be significant. Could it be that these phenomena are (in part) responsible for the apparent enhanced activity or displacement acceleration measured in the 10-month post-failure period?

3) Figure S2. The precipitation record. . . - this figure is important and I suggest moving a possibly modified version from the Supplement to the main article. It could be good to extend the precipitation data to the entire 10-month post-failure period covered by InSAR data.

Minor issues: - Page 2, line 20 "coherence" – might want to explain or at least say interferometric coherence - Page 2, line 29 "over much of the landslide area" – this seems too optimistic if one looks and Figs 7 and 9. - Page 3, Line 13 "up to 50 km2"? Is this correct? - Page 8, Line 15 "(∼2.6 cm. . .)" – Shouldn't it be 1.3 cm?

References

Wasowski J, Bovenga F (2014a) Investigating landslides and unstable slopes with satellite Multi Temporal Interferometry: current issues and future perspectives. En-gineering Geology 174:103–138 http://dx.doi.org/10.1016/j.enggeo.2014.03.003

Wasowski J, Bovenga F (2014b) Remote sensing of landslide motion with emphasis on satellite multitemporal interferometry applications: an overview. In: Shroder JF, Davies T (eds) Landslide hazards, risks and disasters. Elsevier Inc., Amsterdam, Netherlands. (ISBN 978-0-12-396452-6), pp 345–403 http://dx.doi.org/10.101/B978-0-12-396452-6.00011-2

---

## Author Comment (AC1) · 17 Jan 2019

Responses to the referee's comments are provided below. The referee's comments are in bold-faced text. The responses are in plain text with relevant in-text locations provided (underlined text) and added/modified details in the manuscript quotes (red text).

**Anonymous Referee #1**

**Main comments:**

**The presented paper contains very interesting and useful approach and results for landslide movement studies with important implications for the hazard assessment applicable in regions where landslides are major problem. It is very well written and presented and the results are quite interesting for wide audience.**

We have subdivided the referee's description of major issues into five points, below.

1. **But I think that your interpretations are not well supported by the results and available data and that you missed important geomorphological evidences with important implications for the result interpretation. I think that the studied landslide is not suited for investigations of the simple post-failure movement behavior which suggests at least temporal stabiliza tion of the landslide after its major failure.**

   Response: We agree that this is not a simple landslide and that a simple post-failure movement model is inappropriate. Our intention was, in fact, to highlight that such a model, although sometimes applied (including in Bolivia), is not appropriate. The referee's comments indicate that we missed our mark in trying to make this point. We have, therefore, modified the manuscript to better clarify several crucial points (see points 2 to 5 below) including: 1. the complex type and behaviour of this multi-generational landslide; 2. the occasional and informal nature of assumptions of post-failure stabilization; and 3. the concept of post-failure adjustment.

2. **Plus the InSAR results need to be combine with detailed, site specific geomorphological/landslide mapping which was not done (or at least was not presented) and therefore I am convinced that some of the interpretations should be changed/improved.**

   Response: We present general interpretations of the composition and morphology of the slope in Fig. 2, as well as complementary documentation in the text and photo mosaics (Figs. 4 and 5). This overview is based on the only previous mapping of the slope (completed by Anzoleaga et al. [1977] primarily using pre-development aerial photographs) (Fig. 2A) and on our additional observations and interpretations (Fig. 2B). Much of the Pampahasi area is urbanized, with the areas most affected by the 2011 landslide having been quickly modified before the first post-failure imagery. These factors somewhat limit additional interpretations. Although more detailed re-interpretation of pre-development aerial photography might be possible, we feel that such an undertaking is beyond the scope of the current study and could be the subject of a future project. Furthermore, as one of our goals is to demonstrate the utility of HDS-InSAR for

areas with generally limited a priori knowledge of landslides, we feel that the level of background presented for the Pampahasi slope is sufficient for the paper presented here.

We have added/modified text in several locations in the manuscript to clarify the particular importance of advanced InSAR for areas with incomplete details on landslide extent and behaviour:

Abstract: "such details [*InSAR-quantified displacement fields*] are especially useful where knowledge of landslide extent and activity is limited."

Section 2.1 (final paragraph): "Although general geomorphic and geologic characterization has been undertaken for some of the larger landslides (Dobrovolny, 1962; Anzoleaga et al., 1977), detailed site investigations are lacking."

Conclusions (second paragraph): ", and incomplete knowledge of slope activity."

3. **Your statement challenging the post-failure "stabilization" theory are not well documented in the article: You did not defined the "hypothesis" about stabilization after failure - how do you assess the "stability"? I am sure that factor of safety of the main sliding plane calculated after the event would be much higher than before, clearly showing the slope stabilized. Surface movements observed by InSAR technique do not necessarily represent failure plane movements.**

Response: Our mention of assumptions about 'post-failure stabilization' appears to have been problematic because we did not clearly indicate that we meant informal assumptions, typically by non-experts. We have not cited literature about a formal hypothesis for 'post-failure stabilization' because, to our knowledge, none exists for complex reactivated landslides. To remedy this, we have modified text referring to 'post-failure stabilization' as follows:

Abstract: "Changes in deformation in the 10 months following the landslide demonstrate an increase in slope activity and indicate that stress redistribution resulting from the discrete failure has decreased stability of parts of the slope."

Introduction (second paragraph): "Limited post-event monitoring sometimes stems from an assumption that stress release during catastrophic failure enhances slope stability," and "Such records are, however, of utmost importance in light of observed spatiotemporal clustering of failure events (e.g. Hermanns et al., 2006 and references therein; Crosta et al., 2017; Hilger et al., 2018)." (these new sources have been added to the reference list).

Introduction (fourth paragraph): "The Pampahasi case study emphasises the necessity for land-use planning views that better align with the complexity and commonly recurrent nature of large-scale landslides."

Introduction (final paragraph): "Contrary to the sometimes-invoked view that a major failure reduces large-scale instability, the affected slope shows enhanced post-failure activity,

highlighting the complexity of stability and risk for slopes in La Paz, as well as comparable settings."

Section 2.1 (final paragraph): "Land-use decisions in La Paz have historically overlooked the recurrence of slope failures, with large landslide complexes being repeatedly resettled after reactivations or going altogether unrecognized."

Conclusions (first paragraph): "This change in ground deformation counters any expectation that a complex landslide might stabilize, at least temporarily, following a discrete failure and highlights that such an assumption, at least for short-term stability, is imprudent for multi-generational failures. Slope dynamics documented here support observations and theory from scientific literature that spatiotemporal clustering of landslides are responses to stress redistribution, which exceeds stress changes due to background erosion."

Where mentioning 'stability', we are referring to the overall condition of the slope, not only the factor of safety (FoS) of the main failure surface. Even if the FoS for the main failure surface increased as a result of the 2011 reactivation, the entire slope has not necessary stabilized. Our focus in this study is the stability of the overall slope, not just the primary sliding surface. Consequently, we have modified the text to use terms such 'activity', 'overall stability', and 'stability of various parts of the slope', rather than simply 'stability'.

The referee is correct that "*Surface movements observed by InSAR technique do not necessarily represent failure plane movements.*" In fact, one of the benefits of our particular processing methodology is that both extensive, generally deep motion and localized, generally shallow motion can be detected (see text added to sections 5 and 7.2 to address comments from the other referee). In contrast, many existing InSAR techniques preferentially detect spatially regular ground motion resulting from deep-seated failures (cf. Wasowski and Bovenga, 2014, 2015) due to their inability to characterize localized, spatially variable motion. However, some of the enhanced post-failure motion/activity is difficult to explain as only surficial in nature due to its extent and regularity. For example, most of the northern part of the landslide toe moved at a generally similar rate following the 2011 landslide, suggesting involvement of the major failure zone beneath an extensive toe block (see interpretation in section 6.1). We feel that our additions to section 5 and 7.2 – addressing comments from the second referee – also cover the point raised by the first referee.

4. **As far as I know, the post failure "stabilization" has never been the only or major hypothesis about landslide movement. There is very common concept of post-failure adjustment which includes increased activity mainly around the scarp and toe if river is eroding it and locally on sites with steep slopes. Moreover, the landslide you describe represents reactivation of deep and complex landslide and in such cases (regions with long term, complicated landslide history) it has been observed before that reactivation of one part may trigger activity in other areas (domino effect). Whereas I think that the theory of post-failure stabilization has always been limited rather to simple landslide cases (which are not the one you describe).**

Response: We agree with the referee that post-failure stabilization is not the only hypothesis for landslide movement. Our combined experience from work in many parts of the world includes numerous examples in which geoscientist and decision makers have imprudently assumed post-failure stabilization. Recent optical satellite imagery over La Paz, for example, shows that the portions of the Pampahasi slope are being prepared for resettlement. However, we recognize that such views are less common in academic and larger professional settings. We have updated the text in several locations to: 1. specify that assumptions of post-failure stability are sometimes made, but do not constitute a formalized hypothesis (see major point 3); and 2. clarify that the landslide is complex (see major points 2 and 5).

We have included implicit and explicit mention of the concept of post-failure adjustment:

Abstract: "…stress redistribution resulting from the discrete failure decreased stability of parts of the slope."

Section 6.2 (final paragraph): ", which may reflect temporary post-failure adjustment."

We have also included details clarifying that development of the slope progressed despite warns of likely instability provided on the basis of geomorphology, and that new development is ongoing despite repeated failures:

Section 2.2 (final paragraph): "In light of the geomorphic evidence of recurrent instability at the sites, Scanvic and Girault (1989) recommended that this area not be developed. However, the initially sparse development greatly expanded during the last decade of the twentieth century and first decade of the twenty-first, resulting in the establishment of several large neighbourhoods." (this new source has been added to the reference list)

Section 4 (final paragraph): "Large portions of the landslide complex have been recently resettled or are being prepared for reoccupation, with limited control of slope infiltration and runoff."

5. **Therefore I disagree with interpretation of your measurements – they very nicely document complex landslide behavior which correct (as far as I can tell from looking at the Earth Google images) explanation would require detailed geomorphological interpretation of the well morphologically defined "paleo landslide" (yellow line on the attached Fig. 1 EarthGoogle image below), which N limit is some 400 from the 2009 landslide well defined by escarpment. It seems that this paleo landslide body is strongly segmented by gullies running to the main river valley as well as significant slope forming toe of the 2009 landslide above the creeping region with "i" on Fig. 7A.**

Response: The referee's comment indicates that we have not sufficiently communicated what we believe to be nature of landslides constituting the Pampahasi slope. We agree that the Pampahasi landslide is a complex failure consisting of multiple generations of failure and multiple, interconnected components (see also major point 2). To further clarify this, we have

noted in several additional places in the text that the Pampahasi paleolandslides, its most recent phase of activation (the 2011 Pampahasi landslide), and many large ancient landslide in the La Paz area are complex, multicomponent failures. These include:

Abstract: "We characterize and compare creep preceding and following the complex 2011 Pampahasi landslide..."; "The failure remobilised deposits of an ancient complex landslide…"; "…La Paz, half of which is underlain by similar, large and generally complex paleolandslides."

Section 6.2: "The failing slope has many interconnected components that are largely within, but also beyond, the Pampahasi paleolandslide."

Section 8 (second paragraph): "…large-scale reactivations of complex landslides forming these slopes."

The yellow zone annotated by the referee in GoogleEarth (their Fig. 1) encompasses both the Pampahasi paleolandslide and the Villa Salomé paleolandside, which we believe to be a separate landslide mass (see our Fig. 2A). To further clarify our interpretation of two separate landslides that probably act independently of one another, we have added the following text to the final paragraph of section 2.2. "The adjacent and similarly incised Villa Salomé paleolandslide is likely a separate landslide complex because its failure surface is higher and is separated from the Pampahasi paleolandslide, along Río Jankopampa, by a >120-m-thick sequence of intact La Paz Formation (Fig. 2A)."

**Minor issues:**

1. **page 3, line 17-18:** In reference to the statement "Recent landslides, particularly those <1 $Mm^3$, have happened mainly during the monsoon season (December-March),…" the reviewer points out the following: **According to the following sentence** [see minor issue 2, below]**, this statement is not true! I would also suggest to use "rainy/dry seasons" instead of "monsoon" as this phenomena in South America is slightly different compared to the original, Asian one and the term "dry season" is used in the text.**

   Response: This comment and the subsequent one (see 'Minor issue 2' below) show that we did not clearly explain the timing or high frequency of historic landslides in the La Paz area and that we poorly worded some of the text. We agree with the suggestion to use "rainy season" rather than "monsoon season". To address these issues, we have re-written this sentence and the following sentence as follows: "Failures <1 $Mm^3$ in size occur yearly and happen mainly during the rainy season (December-March) (O'Hare and Rivas, 2005; Roberts, 2015). In contrast, of the seven historic landslides larger than 1 $Mm^3$, four happened during the dry season (April-November; Table S1)."

2. **page 3, line 19:** In reference to our statement "Only four of the seven historic landslides larger than 1 $Mm^3$ have happened during the dry season (April-November; Table S1)." the reviewer states: **I think the sentence should be re-phrase - since majority of the historical cases**

**happened during dry season!! Which is, generally, quite "disturbing" finding clearly showing that the landslide occurrence pattern of specific landslide types is much more complex.**

Response: See 'Minor issue 1' above.

3. **page 4, line 16-17:** In reference to the statement "…but whether it [*the Pampahasi paleolandslide*] represents one or multiple events is uncertain." the reviewer states: **There are no satellite or aerial photographs which may help to solve this problem?**

   Response: Aerial photographs (starting in the 1930s) and high-resolution satellite imagery (starting c. 2004) are suitable for depicting modifications/reactivations of the Pampahasi paleolandslide during the twentieth and twenty-first centuries. However, our statement referred to the main components of the landslide, which are prehistoric. To better clarify this, we have modified the text in two places in the fourth and fifth paragraphs of section 2.2:

   "Small (<1 to 20 ha) failures evident in aerial photographs have occurred since at least the early twentieth century within large (~20-200 ha) prehistoric failures in many places."

   "Details of the prehistoric Pampahasi paleolandslide, which substantially predates historic records, must be inferred from its surface expression because no subsurface investigations have been conducted on this slope."

4. **Supplement, page 5, line 1-3: What about the latin? text on the top of the page 5 of your supplement? Some sort of joke? Instead of this I would prefer few lines about how the geotechnical parameters in the second table were derived - lab tests, field measurements, expert estimation?**

   Response: The Latin text was erroneously inserted and has been removed. Directly before Table S2, we have inserted the following summary noting how the geotechnical parameters reported by Anzoleaga et al. (1977) were produced: "Anzoleaga et al. (1977) provide detailed sedimentological and geotechnical characterization of lithostratigraphic units throughout the La Paz area, including units underlying the Pampahasi slope (Table S2). Their data were derived from laboratory testing supplemented by field observations of structure and lithology."

5. **page 4, line 28 (referring to the Supplement): Check the table numbering in the supplement - you have two S2 tables there.**

   Response: The first of the two tables labeled 'Table S2' should have been 'Table S1'. We have corrected this error and reviewed both the main paper and supplement to ensure that cross references to the tables are correct.

6. **page 5, line 20-21:** In reference to the statement "We mapped features of the 2011 landslide from the first cloud-free, postfailure imagery (WorldView-2 acquired on 23 March 2011),…" the reviewer states: **Would be nice to know the occurence day of the 2011 reactivation!**

Response: The date of the 2011 reactivation is provided in section 4, later in the paper. We agree with the reviewer that it would be helpful to also indicate it here. To address this issue, we have updated the text to read: "We mapped features of the 2011 (26 February to 1 March) landslide from the first cloud-free, post-failure imagery (WorldView-2 acquired on 23 March 2011),…"

7. **page 7, line 4-5:** In reference to the statement "It included small volumes of in situ,…" the reviewer stated: **I am not sure what do you mean? In addition to the paleolandslide deposit?**

Response: The reviewer's interpretation is correct. To clarify our meaning, we have modified the text as follows: "In addition to the previously failed material, it included small volumes of in situ,…"

8. **page 7, line 10-13:** In reference to the statement "In response, the Municipality of La Paz increased risk communication and conducted localized remedial engineering works (Hermanns et al., 2012). However, due to their localized and shallow nature, the stabilization efforts probably had little, if any influence on the stability of the slope or behaviour of the subsequent failure." the reviewer stated: **Would be interesting to shortly specify the measures.**

Response: We have clarified the nature of the remedial measures as follows: "In response, the Municipality of La Paz increased risk communication and installed concrete pillars to remediate what became the 2011 landslide headscarp (Hermanns et al., 2012). However, due to their localized and shallow nature, the stabilization efforts probably had little, if any influence on the stability of the slope or behaviour of the subsequent failure."

9. **page 7, line 24:** In reference to the statement "…uplift of several metres at the east end of the bridge suggests a rotational failure zone passing, at least locally, under Río Irpavi." the reviewer commented: **This is not shown on landslide profile on fig. 10.**

Response: We have added this detail to Fig. 10B.

10. **page 8, line 3:** In reference to the statement "Due to its moderate to low velocities and the immediate evacuation of the area,..." the reviewer stated: **You reported velocity of several meters per sec in the scarp region??**

Response: We have updated the text in two places to specify that most, although not all, of the 2011 failure involved rates that were moderate or slower and that residents evacuated following the first evidence of failure (along Río Chujilluncani) from the portion of the landslide the moved fastest:

"Motion was fastest (up to several metres per second) and largely vertical in the head region, which failed shortly after, lasting one to two hours and forming a ~60°, 80-m-high headscarp (Fig. 5A)."

"Due to prompt evacuation of the headscarp area and the dominantly moderate to low movement velocities farther down slope, no lives were lost during the landslide."

11. **page 8, line 10-12:** In reference to the statement "InSAR-measured ground deformation in the Pampahasi area is almost entirely restricted to mapped paleolandslide deposits and the terrain immediately behind their headscarps…" the reviewer stated: **I see singnificant movements N of 2009 landslide and between 2011 and 2009 landslide scarps which I think are not "immediately" behind them. To me it seems as progressive up-slope enlargement (begining of development of other landslides) of the old 2009/2011 landslides as response to the 2011 reactiavation.**

    Response: The areas of ground motion north of the 2009 landslide and between the upper portions of the 2009 and 2011 landslides are not directly behind the scarps of those recent landslides. However, these areas are within or just behind the paleolandslide deposits (denoted in Fig. 2A), which are more extensive. To clarify our intended meaning – the coincidence of these moving areas with paleolandslide deposits, as opposed to the 2009 and 2011 landslides – we have changed the text to:  "InSAR-measured ground deformation in the Pampahasi area is almost entirely restricted to mapped prehistoric landslide deposits, namely the Pampahasi and Villa Salomé paleolandslides (Fig 2A) and the terrain directly behind their headscarps (Fig. 7)."

12. **page 8, line 24-25:** In reference to the statement "There it [*motion north of Río Jankopampa*] terminated abruptly ~50 m north of the river (Fig. 7A) at a locally stable slope comprising undisturbed La Paz Formation." the reviewer stated: **Would be nice to specify this area on the figure as well.**

    Response: Because this figure already contains numerous markers (letters for photo locations [shown in Figs. 4 and 5] and roman numerals for displacement histories [shown in Fig. 8]), we prefer not to add further makers. Instead we have updated Fig. 7A to include the main rivers described in the text, including Río Jankopampa.

13. **page 8, line 28:** In reference to the statement "Pre-failure movement in the upper half of the Pampahasi paleolandslide was localized." the reviewer stated: **Are you sure you have enough data to make this "clear" conclusion? There are large areas with no data and when looking on the fig. 7A, the high movement regions could make linear features across the future source area suggesting development of pre-failure scarps?**

    Response: To ensure that our interpretation is conservative, we have slightly modified the text as follows: "Pre-failure movement in the upper half of the Pampahasi paleolandslide was restricted to an area of no more than ~250 m along slope by ~300 m down slope. The movement pattern is compatible with even more localized activity, but such an interpretation cannot be made with certainty due to presence of several sizable data holes that obscure how much of this area was moving. Given the presence of stationary ground in some parts of the upper landslide (north and south o f 'b' in Fig. 7A, B), creep may have been localized to a few smaller areas."

14. **page 9, line 12-13:** In reference to the statement "The maximum inferred downslope displacement rates following the landslide (14 cm/a; 9 cm/a LOS; Fig. 7D) were similar to those before it, but occurred in a region of the toe that was previously stable ('iii') and along the new headscarp ('vi') where creep had previously been slow (0.5 cm/a LOS)" the reviewer stated: **This is movement pattern which under the presented conditions would be expected representing adjustment of the newly deposited landslide toe (affected stream erosion?) and slope (platform) behind the main scarp (which is very high).**

Response: We have added text to the third paragraph of section 6.1 to acknowledge this interpretation as a possible explanation for increased activity along the landslide toe: "…and may be related to consequent adjustment of stream erosion along the east margin of the deposit."

Existing text in the final paragraph of section 6.1 indicates our agreement with this interpretation for motion behind the head scarp of the 2011 landslide: "Movements on the plateau are likely the result of dilation of the Pampahasi gravel in response to removal of material to a depth of up to 80 m along the 2011 headscarp (Fig. 5A)."

15. **page 9, line 17-18:** In reference to the statement "Detection of movement is not possible over much of the middle and upper parts of the 2011 landslide due to decorrelation resulting from earthworks that continued for years after the event. However, because this zone is bordered on all sides by moving terrain (>1 cm/a LOS; Fig. 7C), it also was likely creeping throughout the period of the post-failure stack." the reviewer stated: **You may use the same manner of interpretatin also for the pre-failure movements on Fig. 7A - see my comments about "localized" movements above. Moreover, this is just a hypothessis, which is not approved by the results for the 2009 landslide - there is high movement around its limits, but id does not continues inside the ladslide (I suppose that the earth works were performed there before 2011 landslide?).**

Response: A similar type of interpolation could be made for the much smaller area of missing data in headscarp area of the 2011 landslide in the pre-failure stack, but only to a degree. Prior to failure, there is a smaller no-data area in the source area surrounded by areas moving at 0.5 cm/a or more, as well as areas showing no movement in the LOS direction. In contrast, in the post-failure stack that the referee has highlighted, the no-data area is surrounded on all sides by a wide (> 150 m) fringe of motion of at least 1 cm/a that gradually decreases in magnitude away from the no-data area.

The more extensive data coverage in the pre-failure stack helps to further inform our interpretations. We have expanded text in the second paragraph of section 5.1 to further clarify our interpretation of the pre-failure pattern (see minor point 13 above).

Yes, this [interpretation of the large area of no data in the post-failure stack] is a hypothesis, but we feel it is a realistic one. We have revised our wording to indicate that much, but not necessarily all, of this area was probably moving: "…much of it also was likely creeping

throughout the period of the post-failure stack." We feel it is unlikely that the entire no-data area is stationary during the period covered by the post-failure stack.

Earthworks at the site of the 2009 landslide were completed before the 2011 failure event. Coherence within the area of the 2009 landslide is thus similarly absent during the pre-failure stack. Coherence in several parts of the 2009 landslide increased in the stack post-dating the 2011 landslide and suggests downslope movement within an area rimmed on most sides by creep. These ground motion patterns are generally similar to post-failure motion around the area of earthworks within in the 2011 landslide. In our opinion, they provide further support for our interpretation of probable creep in the post-failure stack within the large no-data area of the 2011 landslide.

16. **page 10, lines 12-13:** In reference to "…likely drove creep of the 12-ha zone of paleolandslide material directly upslope (to the northwest)." the referee commented: **Please, indicate it on the figure so it is clear to what regions do you refer.**

    Response: Rather than adding further annotation to the figure, we have updated the text to more clearly indicate the area to which we are referring: "…likely drove creep of the 12-ha zone of paleolandslide material ('v') directly upslope (to the northwest)."

17. **page 10, line 15:** In reference to "A 1-ha zone near the midline of the landslide toe ('ii') is one of the few locations of decreased post-failure activity (Fig. 9).", the referee commented: **To me, it seems like very steady movement with only short decrease before and after the landslide? I see no clear slowing down on fig. 9 or 7?**

    Response: This location ('ii') shows a lower rate of displacement in the post-failure stack (Fig. FD) than in the pre-failure stack (Fig. 7C). The change is subtle and not as obvious in Fig. 8A, in part because the HDS point from which the displacement time history comes is not from the zone with the greatest change. The change in most apparent in Fig. 9 (differencing of post-failure and pre-failure linear displacement rates), where it appears as a small blue zone.

18. **page 10, lines 27-28:** In reference to "…suggesting that delayed infrastructure damage there relates to the transition from the failure event to the new post-failure instability regime." the referee commented: **Do I understand is correctly that this landslide part did not move during the event itself?**

    Response: Motion here was sufficiently minor that the exact timing is difficult to determine. The buildings collapsed on February 28 and March 1 (shown in Fig. 5F), indicating that much of the motion occurred during the end of the ~4-day failure period. To clarify these details, we have updated the text as follows: "Creep in this area was greatest directly upslope of the place where several buildings collapsed (Fig. 5F) in the final days (February 28 and March 1) of the four-day landslide event, suggesting that delayed infrastructure damage there relates to the transition from the failure event to the new post-failure instability regime."

19. **page 12, line 20:** In reference to "On-going fluvial down-cutting and toe erosion by Río Irpavi helps to maintain the slope's meta-stable condition.", the referee commented: **How do you explain the very different behaviour of the N and S parts of the landslide toe before the failure? Could it be explained by different river erosion?**

    Response: The precise role of fluvial erosion in during the period of InSAR monitoring is uncertain and cannot be evaluated with confidence using the available data. However, differing river influence on the two parts of the landslide toe is possible. We have added text to explicitly recognize this possibility: "…spatiotemporal changes in fluvial erosion may have contributed to the differing pre-failure activity in the northern and southern parts of the landslide, as well as the increase in post-failure activity in both these areas."

20. **page 13, lines 11-13:** In reference to "or more abrupt acceleration over just a few RADAR acquisitions, but not shorter term changes. Sporadic displacement activity that may signal impending acceleration (cf. Kalaugher et al., 2000) could similarly have gone undocumented." the referee commented: **This fact is not enugh reflected in your results - e.g. Fig. 10C which gives impression of complete displacement record!**

    Response: The referee makes a good point. We have inadvertently overrepresented the temporal regularity of the movement record in this plot. We have removed the unnecessarily precise (i.e. day count and timing of RADAR acquisitions) and changed the solid line to a dashed line. We have also updated the figure caption to explicitly mention the schematic nature of the figure. "C. Schematic temporal displacement history generalized from InSAR measurements bracketing the 2011 Pampahasi landslide and eyewitness accounts."

21. **page 14, lines 17-18:** In reference to "…HDS-InSAR has provided detailed characterization of a large, dynamic urban slope.." the referee commented: **What about effects of fundation/construction quality of houses used as reflectors on the results?**

    Response: Based on field observations as well as the absence of motion and limited phase noise in areas free of landsliding, we conclude that buildings serving as permanent scatterers are stable. Additionally, the exact phase source / target represented by a single pixel is uncertain given the very high density of reflectors relative to the RADAR ground resolution. The relatively large-scale spatial correlation documented here rules out the influence of individual structures. Furthermore, a few settling structures would be given little or no weight during interpolation to produce linear displacement maps. In other words, structure-specific thermal affects and foundation settling will have very short spatial correlation that will not appear in these results. Finally, thermal expansion of common building materials in the Pampahasi area (fired brick and adobe walls; ceramic tile and corrugated metal roofs) are subject to little thermal expansion and should vary little from scene to scene given the inter-seasonal regularity of temperatures in La Paz.

    To address this point, we have included the following text in Section 6.4: "Ground motion represented in displacement maps is independent of the structural behaviour of the built

environment. Isolated building instability is likely in light of some local construction practices in La Paz, but will be extremely localized and thus are removed during spatial interpolation of the maps. Phase change due to thermal expansion will be minimal given limited seasonal temperature differences in the study area. Due to their cyclic nature, any such phase component will not influence long-term displacement trends."

22. **page 14, lines 26-28:** In reference to "This change in ground deformation is counter to the expectation that slopes commonly stabilize, at least temporarily, following a discrete failure and demonstrates that such an assumption, at least for short-term stability, is imprudent for some slopes." the referee commented:  **You can not make this statement due to number of reasons: 1) you did not defined the "hypothessis" about stabilization after failure - how do you assess the "stability"? I am sure that factor of safety calculated after the event would be much higher than before, clearly showing the slope stabilized. Surface movements observed by InSAR technique does not necessairly represent failure plane movements. As far as I know, the post failure "stabilization" has never been the only or major hypothessis about landslide movement. There is very common concept of post-failure adjustment which includes increased activity mainly around the scarp and toe if river is eroding it and locally on sites with steep slopes. Moreover, the landslide you describe represent reactivation of deep and complex landslide and in such cases (regions with long term, complicated landslide history) it has been observed before that reactivation of one part of such a region may trigger acctivity in other areas (domino effect). Whereas I think that the theory of post-failure stabilization has always been limited rather to simple landslide cases (which is not the one you describe).**

Response: Each aspect of the referee's comment is addressed above in our responses to major issues (points 1-5).

23. **page 15, lines 11-12:** In reference to "Improved understanding of instability of the Pampahasi slope is instructive in evaluating and reducing risk from large-scale slope instability in the city of La Paz," the referee commented:  **Despite the improvements, I think you missed an important information - continuous and post-failure accelerated? creep of the northern sector of the paleolandslide (area around "v" on Fig. 7B and N of the 2009 landslide) - from the fig. 2 and much more clearly from the GoogleEarth images, it is clear that this region belongs to the same paleo landslide as the 2011 reactivation thus we may speculate that the 2011 event made this part less stable pointing out possible future "catastorphic" failure location??**

Response: We agree that the area around point 'v' belongs to part of the Pampahasi paleolandslide (Fig. 2A) and that apparent stress redistribution in the slope following the 2011 reactivation may make that zone more susceptible to large-scale rapid failure. Whether the patches of increased post-failure motion north (i.e. upslope) of the 2009 landslide (Fig. 9) are related to the 2011 reactivation of the Pampahasi paleolandslide is less certain. The exact boundary in this area between the Pampahasi and Villa Salomé paleolandslides, which we believe are two separate paleolandslide complexes, is unknown. We have updated the text as follows to clarify this: "Additionally, stress redistribution suggested by accelerated creep of the

Pampahasi paleolandslide between the 2009 and 2011 failures and possibly adjacent parts of the Villa Salomé paleolandslide may increase their susceptibility to large-scale rapid failure in the future."

24. **page 15, lines 13:** In reference to "The Pampahasi slope is similar to other slopes in the city where many of the largest historic landslides have occurred…" the referee commented: **Could you show them  e.g. on Fig. 2?**

Response: These historic failures, as well as the general extent of paleolandslides comprising slopes in the La Paz basin, are shown in Fig. 2B (respectively as purple and light blue polygons). To better clarify this we have updated the text as follows: "The Pampahasi slope is similar to numerous other slopes comprising large ancient landslides in the city, including slopes where many of the largest historic landslides have occurred (Fig. 2B). These slopes are underlain by generally weak, fine-grained sediments of the La Paz Formation."

25. **page 15, lines 17:** In reference to "Evaluating the possibility of future, large-scale reactivations of these slopes requires consideration of high-rainfall scenarios and should not be based solely on creep acceleration." the referee commented: **You forgot about the fact from the introduction that more than 50% of very large landslides happened during the dry season!**

Response: We agree with the referee that variable seasonal timing of the larges landslides should be reemphasised here. However, we also feel that this pattern does not mean that the role of precipitation should be ignored when considering possible future large landslides, particularly since the largest modern landslide (the event described here) directly followed one of the wettest days on record. We have thus modified the text as follows: "Evaluating the possibility of future, large-scale reactivations of complex landslides should not be based solely on creep acceleration. Although fewer than half of the historic failures exceeding 1 Mm$^3$ happened during the rainy season, coincidence of the 2011 reactivation of the Pampahasi paleolandslide with particularly wet conditions indicates that consideration of high-rainfall scenarios is advisable."

26. **Figure 7:** The reviewer stated: **For this figure, I would suggest to limit the pre-failure period also to only 10 months since the displacement histories on Fig. 8 show variable velocities during the observation time. Using the same temporal window the results would be better commparable.**

Response: Limiting the temporal coverage of the pre-failure stack to the 10 months preceding the 2011 landslide would reduce the precision of that stack, but make it more comparable in quality to the post-failure stack.

We prefer not to take this approach for two reasons. First, we believe that it is more important to limit, as much as possible, sources of error in the InSAR data. Thinning of the pre-failure stack would decrease precision of that dataset by roughly half (see various points about data precision estimation in the responses to the second referee's). Additionally, the extent of nodata areas and the degree of phase artifacts would undoubtedly increase. Second, given the seasonality of precipitation in the Pampahasi area, considering two different month ranges (i.e. May to February for the pre-failure stack; March to December for the post-failure stack) will not necessarily increase the comparability of the two datasets.

27. **Figure 9 caption:** The reviewer commented: **Please, specify dates of the analyzed images.**

Response: We have updated the caption to indicate the imagery used for each of the stacks as follows: "Figure 9. Change in line-of-sight slope creep of the Pampahasi area following the 2011 failure relative to creep before failure. Positive and negative values indicate, respectively, decreased and increased post-failure displacement rates away from the satellite. Green areas experienced no change in displacement and were largely stable both before and after the 2011 landslide. White areas lacked coherence phase in pre-failure scenes, post-failure scenes, or both. Roman numerals show locations of displacement histories in Figure 5. Pre-failure and post-failure imagery spans, respectively, 08 September 2008 to 13 February 2011 and 09 March 2011 to 22 December 2011."

---

## Author Comment (AC2) · 17 Jan 2019

Responses to the referee's comments are provided below. The referee's comments are in bold-faced text. The responses are in plain text with relevant in-text locations provided (underlined text) and added/modified details in the manuscript quotes (red text).

**Anonymous Referee #2**

**Main comments:**

1. **Data quality/relevance - The authors discuss some limitations of InSAR data and mention that the quality of the post-failure InSAR results is expected to be lower than that obtained from the thinner stack of the pre-failure radar acquisitions. This is correct, but since you compare the InSAR results from two different periods, I think the issue of data quality deserves more attention.**

   The points raised by the referee are amplified and addressed below:

   1.1. **The precision of InSAR measurements depends also on the environmental conditions and the adopted coherence threshold (cf. Wasowski and Bovenga, 2014a,b). For instance, after the Feb 2011 failure, the topography of the slope has changed with respect to the reference SRTM DEM. Significant change for InSAR processing and sensitivity to displacement? Representative prefailure and post-failure topographic sections of the landslide could help. Also, was the weather less rainy (dry) in the post-failure period with respect to the pre-failure time? Significant for the processing results?**

   Response: We agree that the three sources – topography, precipitation / soil moisture, and coherence threshold – noted by the referee as having possible influences on InSAR precision are worth explicitly evaluating. To address these three possible sources of differential error between the pre-failure and post-failure stacks we have added: 1. two paragraphs in section 6.4 ('Sources of uncertainty') in the main body of the paper; 2. text in the captions of Figs. 8 and 10; 3. two display items to the Supplement; and 4. several lines of text to the Supplement.

   *Additions to main paper*:

   We have added two paragraphs to section 6.4 to provide a conservative estimate of InSAR precision and the limited role of topographic and moisture change: "The HDS-InSAR processing chain is complex and includes many non-linear steps, which greatly complicates development of an accurate error approximation model. Both the pre-failure and post-failure stacks have good baseline diversity, allowing relative errors between them to be approximated by first-order estimates from the square root of the number of scenes (32 vs. 12). In the absence of a rigorous model, we assume that error is conservatively as large as twice for the thinner stack. We approximate the errors as 3 mm/a and 6 mm/a, respectively, for the pre-failure and post-failure stacks.

   Due to the structure of the HDS-InSAR processing chain, differing environmental conditions between the two stacks – namely topography and moisture (cf. Wasowski and Bovenga, 2014,

2015) – have minimal effects. The reference digital terrain model (DTM) is used only for an initial topographic correction; stack processing solves for height error relative to the reference DTM and provides a new elevation solution for each of the two stacks (Supplement), which improves terrain representation. Topographic correction of the post-failure stack thus accounts for landslide-induced terrain changes, which were greatest in the source area (Fig. 10B). Temporal soil moisture variability is unlikely to affect phase by more than 100° (Rabus et al., 2010), which equates to approximately one-third of an interferometric fringe or 0.9 cm for the sensor used here. Comparison of precipitation records in the 30 months before and 10 months after the 2011 Pampahasi landslide indicates that long-term precipitation amounts during the pre-failure and post-failure stacks were comparable. Spatial moisture gradients are a more substantial error source (Rabus et al., 2010), but major differences in a single scene are exotic events that are removed during stack processing."

The relevant references (Wasowski and Bovenga, 2014, 2015; Rabus et al., 2010) have been added to the reference list:

"Rabus, B., When, H., and Nolan, M.: The importance of soil moisture and soil structure for InSAR phase and backscatter, as determined by FDTD modeling, IEEE T. Geosci. Remote, 48, 2421–2429, doi: 10.1109/TGRS.2009.2039353, 2010."

"Wasowski, J., and Bovenga, F.: Investigating landslides and unstable slopes with satellite Multi Temporal Interferometry: Current issues and future perspectives, Engineering Geology, 174, 103–138, http://dx.doi.org/10.1016/j.enggeo.2014.03.003, 2014."

"Wasowski, J., and Bovenga, F.: Remote sensing of landslide motion with emphasis on satellite multitemporal interferometry applications: an overview, in: Davies, T. and Shroder, J.F. (Eds.), Landslides Hazards, Risks and Disasters, Academic Press, Amsterdam, Netherlands, 345-403, https://doi.org/10.1016/B978-0-12-396452-6.00011-2, 2015."

We have added a sentence to the subsequent paragraph of section 6.4, which discusses the approximation of 3D displacement from RADAR line-of-sight displacements, to reflect additional error represented in the approximation of the full motion vector from LOS motion: "Additional error is thus introduced in the conversion from measure one-dimensional (LOS) displacement to approximate three-dimensional (true) motion."

As suggested by the reviewer, Fig. S2 from the Supplement of the original submission will be moved to the main paper. We prefer not to expand panel B of this figure to cover the entire 10-month post-failure period of InSAR coverage. Such expansion would greatly compress the time axis of the plot, making antecedent conditions in the month leading up to the 2011 Pamaphasi landslide difficult for the reader to decipher. We have instead added two display items to the Supplement (see below).

We have added a sentence to the caption of Fig. 8 to reflect additional error represented in the approximation of the full motion vector from LOS motion in panel B: "The right axis is

approximate, given that additional error is introduced from conversion from LOS to true motion."

We have added a sentence to the caption of Fig. 10 to further clarify the degree of topographic changes due to the 2011 landslide event and thus the magnitude of difference between post-failure terrain and the pre-failure reference DTM: "…, which depicts the generalized pre-failure terrain. Dashed and solid line work in the source area (0 to ~150 m x-axis distance) represents, respectively, pre-failure topography and post-failure topography based on terrain changes measured in the field using a handheld laser rangefinder. Terrain lower on the failed slope changed comparatively little during the landslide; solid line work beyond ~150 m x-axis distance is thus generally representative of both the pre-failure and post-failure topography."

*Additions to Supplement*:

A table and a figure will be added to the Supplement to show that precipitation during the 10-month post-failure window (March through December 2011) was generally characteristic of the same months in the pre-failure window (March through December in each of 2008, 2009, and 2010). These new display items are now referenced in the discussion of the main paper in a brief comparison of pre-failure and post-failure precipitation conditions (see 'Additions to section 6.4' above).

The two additional supplemental display items noted above are complemented in the Supplement by a brief description of pre-failure and post-failure precipitation conditions: "Despite the high variability of short-term precipitation evident from eyewitness accounts of rainfall intensity and from daily precipitation records, the longer term (monthly to annual) amount of precipitation before and after the 2011 landslide was similar. Precipitation during the 10-month post-failure period (March to December, 2012) totalled 282.2 mm and ranged on a monthly basis from 0 to 127.6 mm. These conditions are typical of March-to-December precipitation in each of the three years fully covered by the pre-failure InSAR stack (i.e. 2008, 2009, and 2010: 0 to 137 mm monthly; 235 to 366 mm 10-month total; 282 mm 10-month mean)."

We have added a sentence in the processing overview provided in the supplement to specify the stack thickness beyond which statistics are more robust: "The HDS-InSAR processing approach generates stronger statistics for stacks of approximately 15 or more scenes."

**1.2. Fig 6b, which shows the distribution and average annual velocity of HDS (measurement points) for the pre-failure period, indicates some "noise" (especially southern part) in the data (areas where different velocity points are mixed together). Perhaps it would be useful to show a similar figure for the post-failure period. If not, to give an idea of the precision of the results obtained for the pre- and post-failure, you could consider estimating the mean velocity standard deviations for the two periods (cf. Wasowski and Bovenga, 2014a,b).**

Response: Showing similar graphics (HDS points and interpolated linear deformation map) for the post-failure period is an excellent suggestion. Instead of creating a new, separate figure, we have opted to modify Fig. 6 to include these post-failure graphics for the sample area. Representing both the pre-failure and post-failure stacks in this single figure parallels the structure of Fig. 7 and will allow the reader to directly compare results from the two InSAR stacks. The updated figure is further supported by several text modifications outlined below:

We have added text to the first paragraph of section 5 to clarify the sources of noise in HDS and how these have minimal impact on the interpolated linear deformation maps used to analyse slope activity in the Pampahasi area: "Furthermore, the interpolation weighting suppresses very localized HDS clusters that differ greatly from the average (Supplement). These small-scale variations – whether representing surficial movement or noise from uncorrected phase unwrapping errors – are consequently inconsequential to large-scale patterns described below."

Figure 6 caption: We have updated this caption to reflect the addition of the post-failure HDS results and the linear deformation map, we have added text to highlight the noise in HDS results and its reduction the linear deformation maps: "Localized variably in HDS results (panels A and B) is supressed by linear deformation interpolation (panels C and D)."

We have added text to section 4.7 of the supplement to clarify how the interpolation suppressions of very localized variations of HDS values: "Small HDS clusters (fewer than five points) receive no weight during interpolation, resulting in suppression of much of the localised (smaller than several metres by several metres in the case of the RADAR resolution we processed) variations that might be noise resulting from phase unwrapping errors."

In light of the expansion of Fig. 6, we have opted not to estimate mean velocity standard deviations for the pre-failure and post-failure periods (citing Wasowski and Bovenga, 2014a, b) as a way to indicate the precision of the two stacks.

**1.3. Finally, we know the InSAR are relative in both time and space. You indicate the Master scene in the Supplement (for both pre- and post-failure stacks), but the location of a reference point (area) in space is not specified, unless I missed it. This could be relevant considering the generally marginal (changing?) stability of the land in the study area.**

Response: Our method does not have a spatial reference point, but rather uses a commutative reference area made up of all areas detected as non-moving at scales <10 km; movement on scales larger than 10 km are removed during atmospheric filtering. Quantified motion is thus not absolute, but rather relative to this 'non-moving' background area. Consequently, the question of a specific spatial reference is not relevant. The reviewer's comment highlights that we had not effectively conveyed this aspect of the processing. We have made the following additions to remedy this:

Methods additions:

The first sentence of section 4.5 in the supplement has been expanded to read: "We applied differential InSAR (D-InSAR) to determine time-series phase statistics and to identify phase differences on scales greater than ~10 km, which are presumed to be atmospheric phase contributions."

We have added a sentence to the final paragraph of section 4.6 in the supplement specifying use of a cumulative reference area as follows: "Instead of a single spatial reference point or region, phase changes in HDS-InSAR are determined relative to a cumulative reference area comprising all areas lacking movement on scales <10 km, with broader scale motion having been removed by atmospheric filtering."

2. **Interpretations and conclusions - I can follow you for the most part, but remain somewhat uncertain about the postulated broad significance of the observed postfailure creep acceleration (enhanced activity).**

The various points on this topic made by the referee are addressed below (points 2.1 to 2.2):

2.1. **One reason is that the quality/precision of pre- and post-failure measurements is not the same and perhaps difficult to assess (cf point above).**

Response: We agree that the exact quality of the both pre-failure and post-failure InSAR stacks is difficult to assess. However, our conservative estimates of their precisions as, respectively, 3 mm/a and 6 mm/a (see point 1.1 above) are below the long-term displacement rates across much of the Pampahasi area. We have added additional text to the end of the second paragraph of section 6.4 to clarify this: "Because the average displacement rates across much of the Pampahasi area exceeds the simplified error estimates, the displacement patterns documented here are generally reliable."

2.2. **Then, the conclusion regarding the post-failure creep acceleration is based just on this one specific case. On the basis of the literature review and their own data, Wasowski and Bovenga (2014a,b) indicated that InSAR seem to preferentially capture creep of deep slides, seasonal accelerations of large landslides and post-failure ground instability (settlements, volumetric changes). For some deep landslides/materials, these settlements and volumetric changes can be significant. Could it be that these phenomena are (in part) responsible for the apparent enhanced activity or displacement acceleration measured in the 10-month post-failure period?**

Response: Benefits of HDS-InSAR include its improved preservation of spatial resolution compared to other InSAR techniques and its optimization for spatially uncorrelated ground motion (described in section 3.2). These aspects make it particularly well suited for characterizing high spatial frequency ground motion and thus spatially irregular displacement patterns generated by localized landsliding (in both the presence and absence of more regular, large-scale displacements patterns resulting from deep-seated landslides). Consequently, the

displacement maps we present are less biased to large, deep-seated landslides than are maps produced using conventional InSAR methods. To clarify this, we have added the following:

Section 5 (end of first paragraph: "The displacement maps record deep spatially regular slope movements as well as shallower more variable movements, but their differentiation requires consideration of displacement patterns and may not always be clear."

Section 7.2 (second paragraph): "Furthermore, such techniques enable more comprehensive detection and characterization of landslides of different depth and size. Preferential detection of deep landslides by InSAR (Wasowski and Bovenga, 2014, 2015) reflects the typically large spatial regularity of their displacements. Shallow instability, especially in areas of variable micro-topography, generates spatially irregular ground motion that is more difficult to detect. HDS-InSAR's optimization for locally variable ground motion particularly improves the characterization of shallow landslides and thus reduces biasing toward deep-seated instability."

By 'settlement' we presume that the reviewer means post-failure compaction of the debris and thus are questioning whether the enhanced ground displacements in the post-failure stack could be the result of compaction, rather than displacement along failure zones/surfaces. Compaction may be possible in some places. However, much of the area involved in the 2011 event (particularly areas that maintain coherence in the post-failure stack) underwent minimal transport and thus minimal bulking. Consequently, very minimal compaction is possible and the majority of the surface movements most likely reflect true landslide displacements. To clarify this, we have added text to section 5.2 (second paragraph): "Given the limited transport, and thus bulking, across most of the area of the 2011 landslide (Fig. 2A) and the occurrence of abundant surface displacement beyond it limits (Fig. 2B), ground motions following the event should largely represent mass movements as opposed to soil settlement or compaction."

3. **Figure S2. The precipitation record… - this figure is important and I suggest moving a possibly modified version from the Supplement to the main article. It could be good to extend the precipitation data to the entire 10-month post-failure period covered by InSAR data.**

    Response: We have moved this figure from the Supplement to the main paper. Expanding panel B of this figure to cover the entire 10-month post-failure period of InSAR coverage would greatly compress the time axis of the plot, hindering display of antecedent conditions in the month leading up to the 2011 Pamaphasi landslide. To quantify both pre-failure and post-failure precipitation, we have also added the display items mentioned in point 1 above to the Supplement, which will clearly represent precipitation during the 10-month post-failure window (March through December 2011) as well as precipitation during the pre-failure period (2008-2010).

**Minor issues:**

4. **page 2, line 20: "coherence" – might want to explain or at least say interferometric coherence.**

Response: This is a good point and adds clarity. The text now specifies: "…interferometric coherence."

5. **page 2, line 29: "over much of the landslide area" – this seems too optimistic if one looks and Figs 7 and 9.**

Response: We have modified to text to better reflect that displacement fields in some areas, particularly in the post-failure InSAR stack, could not be quantified interferometrically: "…over large portions of the landslide…".

6. **page 3, line 13: "up to 50 km"? Is this correct?.**

Response: The value of 50 km$^2$ is correct. The end-Pleistocene Achocalla earth flow (southernmost landslide deposit in Figure 1B) is the largest landslide in the La Paz area. This failure produced the Achocalla basin directly south of La Paz, which is floored by less mobile, proximal debris (~40 km$^2$). More mobile, distal debris flowed over 20 km down Río La Paz, producing a long debris tongue (~10 km$^2$). We have updated the text to direct the interested reader to a short report on this specific landslide (Dobrovolny 1968), which provides a general description of the landslide and conservatively estimates the volume of the failure material as 2.7 km$^3$. "…including deposits of large (up to 50 km$^2$; Dobrovolny, 1968) paleolandslides…"

The relevant reference has been added to the reference list: "Dobrovolny, E.: A postglacial mudflow of large volume in the La Paz valley, Bolivia. US Geological Survey Professional Paper 600-C, 130–134, 1968."

7. **Page 8, line 15: "(~2.6 cm:…)" – Shouldn't it be 1.3 cm?.**

Response: The value '2.6. cm' was a typographic error. It should be '2.8 cm'. Aliasing will occur when ground motion between successive RADAR acquisitions causes a shift exceeding the sensor's wavelength. Because the two-way travel of waves must be considered (incident beam and reflected beam), the line-of-site threshold for producing temporal aliasing is one-half of the sensor's wavelength (5.6 cm for RADARSAT-2). We have corrected the value and added additional text so that the sentence reads: "…from aliasing where displacement rates are greater than the detection threshold of RADARSAT-2 (~2.8 cm LOS [equivalent to a two-way travel distance for RADARSAT-2's 5.6-cm wavelength] over 24 days)."

---

## Author Response (AR1)

The uploaded manuscript and supplement contain all revisions proposed in response to the reviewer's comments.

Please note that a typographic error has been corrected in the title (addition of a comma after 'Bolivia'). This will need to also be adjusted in NHESS's record for this manuscript.

---

## Author Response (AR2)

No further revisions to the manuscript were required.

Please note the following for final production:

1. A typographic error in the title has been corrected in the files uploaded here; a comma has been added directly after 'Bolivia'. This may also need to be changed in NHESS's document/tacking system.
2. A relevant, very recently published paper has been added to the references and in text citations (in two placed), but the text has not otherwise been altered. In both cases, this citation was added to existing lists of citations.
3. Line work in some figures had to be very slightly adjusted to ensure they confirmed to the journal's formatting requirements (particularly page width sizes).
4. No specific 'Key Figure' has been prepared as this was not mentioned or required at any earlier stage of review. Figure 8 is probably the most appropriate figure for this purpose, but does not 'stand alone' as it includes referenced to other figures. If necessary, we can produce a proper 'key figure' for the final printing.